# LILO: Bayesian Optimization with Natural Language Feedback

Katarzyna Kobalczyk [1]   Zhiyuan Jerry Lin [2]   Benjamin Letham [2]   Zhuokai Zhao [2]
Maximilian Balandat [2]   Eytan Bakshy [2]

## Abstract

Many real-world optimization problems are guided by complex, subjective preferences that are difficult to express as explicit closed-form objectives. In response, we introduce Language-in-the-Loop Optimization (LILO), a Bayesian optimization (BO) framework that employs a large language model (LLM) to translate free-form natural language feedback and prior knowledge from a decision maker into structured preference signals, going beyond the restrictive scalar or pairwise feedback formats typically assumed in preferential BO. The LLM-derived preferences are integrated by a Gaussian process proxy model, enabling principled acquisition-driven exploration with calibrated uncertainty. By placing the LLM in a supporting role rather than as the optimizer itself, LILO preserves the sample efficiency and stability of BO while providing a flexible and expressive feedback interface. Across synthetic and real-world benchmarks, LILO consistently outperforms both conventional preference-based BO methods and LLM-only optimizers, with particularly strong gains in feedback-limited regimes. The code for reproducing our experimental results is available at https://github.com/facebookresearch/lilo.

## 1. Introduction

Bayesian optimization (BO) is a powerful approach for optimizing expensive-to-evaluate black-box objectives (Brochu et al., 2010; Shahriari et al., 2015; Frazier, 2018). In many real deployments, however, the true objective we ultimately care about is not a directly measurable scalar: it reflects a decision maker's (DM's) preferences over complex trade-offs among multiple outcomes. As a result, optimization

[1]DAMTP, University of Cambridge. Work done while the author was an intern at Meta. [2]Meta. Correspondence to: Katarzyna Kobalczyk <knk25@cam.ac.uk>.

*Proceedings of the 43rd International Conference on Machine Learning*, Seoul, South Korea. PMLR 306, 2026. Copyright 2026 by the author(s).

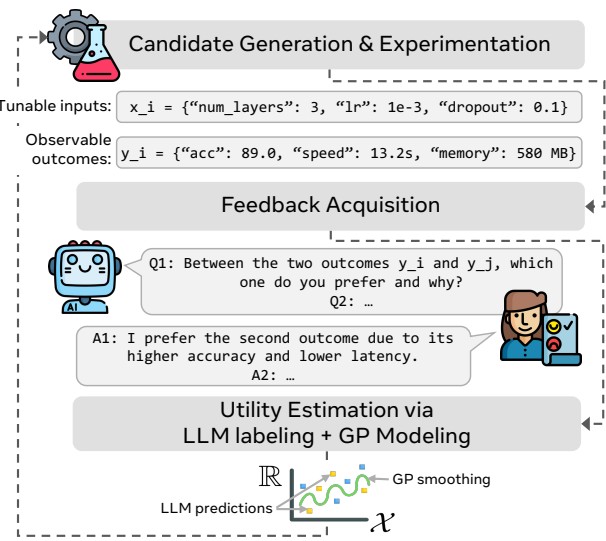

*Figure 1.* **LILO at a glance.** At each BO trial, a surrogate model is used to select a batch of candidate points to evaluate their outcomes. The DM provides free-form feedback messages, and an LLM agent translates the accumulated language into labeled pairwise preferences among observed outcomes, which are used to update the pairwise GP surrogate for the next acquisition step.

must be guided by DM feedback rather than by repeated evaluations of a known utility function.

Existing preference-based BO methods learn from structured feedback such as pairwise comparisons or ratings (González et al., 2017; Chu & Ghahramani, 2005; Lin et al., 2022; Feng et al., 2024). While principled, these approaches assume narrow feedback formats or require problem-specific modeling choices (e.g., custom likelihoods, kernels, or priors) to capture preferences over high-dimensional outcomes. In contrast, large language models (LLMs) provide a flexible interface: they can interpret rich natural language feedback and leverage domain knowledge expressed in text (Brown et al., 2020; Yang et al., 2023; Liu et al., 2024). But when used end-to-end as optimizers, LLM-driven methods generally lack calibrated uncertainty, principled exploration–exploitation tradeoffs, and sample efficiency – properties central to BO.

We introduce *Language-in-the-Loop Optimization* (LILO), a framework that combines the expressivity of language feedback with the reliability of BO. LILO places the LLM

in a *supporting* role: it translates free-form feedback and textual priors into structured preference information, which is integrated by a Gaussian process (GP) proxy model. After each round of experimentation, `LILO` uses accumulated natural language feedback to infer pairwise preferences between observed outcomes, fit a pairwise GP surrogate, and apply standard BO acquisition functions to propose the next batch of candidates. This design preserves BO's calibrated uncertainty and acquisition-driven exploration, while enabling a natural, information-dense feedback channel.

Beyond introducing this hybrid design, we study the key choices that make `LILO` effective in practice, including (i) translating natural language into optimization-relevant utility signals, (ii) performing LLM-based preference labeling under a budget, and (iii) incorporating textual prior knowledge to warm-start the search. Across synthetic and real-world benchmarks, `LILO` consistently outperforms both LLM-based optimizers and BO baselines relying on quantitative feedback, particularly in early optimization stages. In summary, our main contributions are:

1. We propose `LILO`, a Bayesian optimization framework that elicits utility information via free-form natural language feedback, enabling a more expressive interface than standard pairwise or scalar feedback.

2. We introduce a utility estimation mechanism based on scalable LLM-based preference labeling and GP modeling, enabling calibrated uncertainty and acquisition-driven exploration under noisy, indirect feedback.

3. We demonstrate that `LILO` outperforms conventional BO baselines and LLM-only black-box optimization methods across synthetic and real-world problems.

4. We show that `LILO` can incorporate textual prior knowledge to warm-start optimization by leveraging the LLM's pretrained domain knowledge.

5. We also illustrate that `LILO` is naturally compatible with high-dimensional and unstructured outcomes such as text or images.

### 1.1. Problem Definition

We consider the problem of optimizing the outputs of a black-box system with respect to preferences of a DM. Let $\mathcal{X} \subset \mathbb{R}^d$ denote the search space of tunable parameters, and let each $y \in \mathcal{Y} \subset \mathbb{R}^k$ define the outcomes of an experiment obtained through an expensive-to-evaluate, black-box function $f : \mathcal{X} \to \mathcal{Y}, x \mapsto y = f(x)$.

A DM associates with each observed outcome $y$ a latent utility value $u = g(y) = (g \circ f)(x) \in \mathbb{R}$, where $g : \mathcal{Y} \to \mathbb{R}$ is an unknown utility function that reflects the DM's preferences. In practice, the DM is often unable to directly specify the closed form of the utility function, so the exact scalar utility value is not directly observable. Instead, the DM

can provide feedback, e.g., based on pairwise comparisons (Lin et al., 2022) or, like in this work, natural language, to inform about the shape of the utility function $g$. Our goal is to identify parameters $x \in \mathcal{X}$ that maximize satisfaction of the DM, i.e., to solve the composite optimization problem defined as:

$$x^* := \arg\max_{x \in \mathcal{X}} \; g(f(x)).$$

For example, as illustrated in Figure 1, in hyperparameter tuning for machine learning models, $x$ would be the collection of hyperparameters such as batch size or learning rate and $y$ would be the model's evaluation metrics such as accuracy, training time, and memory usage. Given a specific set of outcomes $y$, the utility $u$ reflects the DM's overall satisfaction about the observed outcomes $y$.

Note, this is a highly flexible setup that can accommodate a wide variety of problems. When there is no intermediate outcome that can be observed and the preference is directly defined over the parameter space $\mathcal{X}$, the setting is equivalent to letting $y = f(x) = x$, reducing this problem to the classic *preferential BO* (PBO) problem (González et al., 2017; Astudillo et al., 2023).

As will be shown in Section 4, our proposed solution only models the composite function $g \circ f$, thus, enabling setups where the observable outcomes are high-dimensional and unstructured (e.g. text or images).

## 2. Related Work

**Preferential Bayesian Optimization.** Classic BO methods (Shahriari et al., 2015; Frazier, 2018) combine a probabilistic surrogate model, typically a GP, with an acquisition function that balances exploration and exploitation. Preference learning extends this paradigm to settings where numerical utilities are unavailable and only relative judgments can be obtained from a DM. Early work by Chu & Ghahramani (2005) introduced GP-based preference learning with a probit likelihood for noisy pairwise comparisons, while González et al. (2017) formalized preferential BO and established convergence guarantees. Subsequent extensions explored other querying strategies, including eliciting preferences over hypothetical outcomes (Lin et al., 2022) and best-of-$k$ or multi-dueling formulations (Sui et al., 2017; Astudillo et al., 2023). Together, these methods provide a principled foundation for preference-guided optimization, but they rely on rigidly structured feedback. In practice, however, DMs may want to express their preferences in natural language, conveying high-level goals, tradeoffs, and qualitative constraints that are difficult to capture through conventional feedback formats. In contrast to these works, `LILO` leverages free-form natural language feedback as its primary interface, while retaining a GP surrogate and acquisition-driven optimization over continuous search spaces.

**Optimization with LLMs.** Recent work has shown that LLMs can be excellent few-shot learners and Bayesian predictors (Brown et al., 2020; Panwar et al., 2023; Falck et al., 2024; Requeima et al., 2024; Zhu & Griffiths, 2024). Researchers then explored using LLMs as black-box optimizers by leveraging their in-context learning abilities (Yang et al., 2023). Several studies investigate the use of LLMs for warm-starting, candidate proposal or proxy modeling itself (Liu et al., 2024). Other works explore the synergy of LLMs and BO based on GP surrogates either by employing LLMs to guide exploration (Ramos et al., 2023; Cai et al., 2025; Agarwal et al., 2025) or use them as fixed feature extractors for BO surrogates (Kristiadi et al., 2024). Other approaches assume a discrete set of candidate options represented by predefined feature vectors and use LLMs to elicit preferences over this finite domain (Austin et al., 2024; Handa et al., 2024). While these methods demonstrate that LLMs can be useful in black-box optimization, they often treat the LLM and the surrogate model as parallel or loosely coupled components, or rely on assumptions (e.g., discrete candidate sets or domain featurization) that limit scalability to complex search spaces. In contrast, LILO integrates the LLM and the BO surrogate in an interdependent manner: the LLM's role is to interpret language feedback and generate structured preference labels, which are then absorbed by a GP proxy model governing exploration and exploitation.

See Appendix E for a comprehensive overview positioning LILO with respect to the existing families of related work.

## 3. Background

In this section, we familiarize the reader with pairwise GP models and the EUBO acquisition function which play a key role in preferential BO and our framework.

**Pairwise GP.** A standard model for pairwise preference data follows the work of Chu & Ghahramani (2005), where a latent utility function $g : \mathcal{Y} \to \mathbb{R}$ is endowed with a GP prior. Given a pair of outcomes $(y, y')$ and a binary label $s \in \{0, 1\}$ indicating whether $y$ is preferred over $y'$, preferences are assumed to arise from utility differences modeled with the probit likelihood:

$$\Pr(s = 1 \mid g) = \Phi\left(\frac{g(y) - g(y')}{\sqrt{2}\lambda}\right), \quad (1)$$

where $\Phi$ is the standard normal CDF and the learned hyperparameter $\lambda$ governs the noise of the preferential response. Because the probit likelihood is non-Gaussian, the posterior over latent utilities is intractable. The Laplace approximation is used to obtain a Gaussian approximation. This enables tractable prediction and acquisition optimization. Further details can be found in Chu & Ghahramani (2005).

In this work, we model the composite utility $u(x) = g(f(x))$ as a GP over $\mathcal{X}$; preferences are elicited over out-

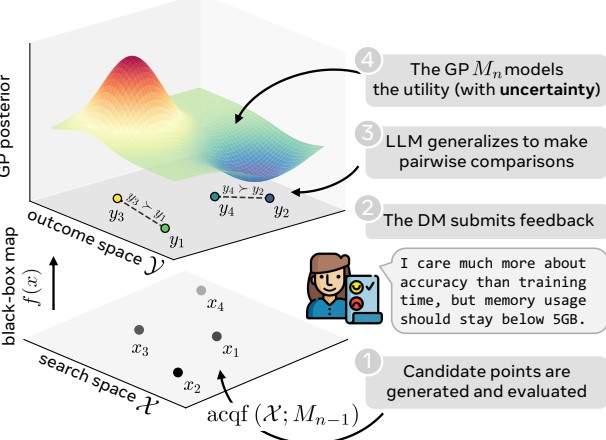

*Figure 2.* **A single natural-language message can constrain many preferences.** The DM's feedback expresses global tradeoffs and constraints on the latent utility $u(x) = g(f(x))$, inducing pairwise preferences among observed outcomes. LILO uses an LLM to convert the feedback history into pairwise labels, fits a pairwise GP proxy $M_n$ to obtain a smooth posterior over utility $u$ with uncertainty for the next acquisition.

comes $y$ but induce pairwise comparisons between the corresponding inputs $x$ so that we are able to optimize and generate new candidates directly in the parameter space.

**EUBO.** In the setup of Lin et al. (2022), the goal is to learn the latent utility function $g$ via obtaining pairwise preference feedback. The Expected Utility of Best Option (EUBO) acquisition function guides preference-exploration by selecting comparisons expected to yield the greatest increase in the utility of the best option. Concretely, consider a pair of outcomes $\{y_1, y_2\}$. EUBO is defined as:

$$\mathrm{EUBO}(y_1, y_2) = \mathbb{E}\big[\max\{g(y_1), g(y_2)\}\big]. \quad (2)$$

Under a Gaussian posterior, the above expression can be computed analytically by standard results for a truncated normal differences. By selecting a pair of outcomes with the highest EUBO, one actively queries about top-utility options, accelerating convergence towards high-utility outcomes while minimizing the number of preference queries. Astudillo et al. (2023) generalize EUBO to $q$ outcomes.

## 4. Method

The key idea behind LILO is to adopt an iterative BO-style algorithm with an explicit GP-based utility model to steer the optimization in a principled fashion while utilizing LLMs to elicit free-form language feedback from the DM and translate it to a numeric preference signal over the observed outcomes. In this section, we describe the core steps of LILO. Algorithm 1 presents the main pseudo-code, while the exact prompt formats used by the subroutines can be found in Appendix B.2.

Our BO procedure consists of $T$ sequential trials indexed by $n \in \{1, 2, \ldots, T\}$. We define by $D_n^{\text{exp}} = \{(x_i, y_i)\}$ the accumulated *experimental dataset* consisting of input–output pairs observed up to trial $n$. Additionally, $D_n^{\text{pf}} = \{(q_j, a_j)\}$ denotes the *preference feedback dataset* that contains a history of the DM's natural language feedback in the form of answers $a_j$ to LLM-generated queries $q_j$. From these datasets, in each trial, we fit a proxy GP model $M_n : \mathcal{X} \to \mathcal{P}(\mathbb{R})$, approximating the composite function $g \circ f$, enabling principled candidate generation in the subsequent trial.

**The entry point.** Before entering the main optimization loop, the algorithm begins by eliciting the DM's high-level optimization goals. At this stage, no experimental outcomes exist ($D_0^{\text{exp}} = \varnothing$), so the LILO agent generates $B^{\text{pf}}$ general questions for the DM. After obtaining the DM's answers, the preference feedback dataset is instantiated as $D_0^{\text{pf}} = \{(q_j, a_j)\}_{j=1}^{B^{\text{pf}}}$. Next, LILO proceeds to the main optimization loop. As illustrated in Figure 1, each trial consists of three stages: (i) Candidate Generation & Experimentation, (ii) Feedback Acquisition, and (iii) Utility Estimation via LLM Labeling + GP Modeling.

---

**Algorithm 1:** LILO

---

**Require:** Max number of iterations $T$, experiment batch size $B^{\text{exp}}$, feedback batch size $B^{\text{pf}}$. **Optional:** Prior knowledge prompt $P_{\text{prior}}$
$D_0^{\text{exp}} \leftarrow \varnothing$;
$\{q_j\}_{j=1}^{B^{\text{pf}}} \leftarrow \text{LILO.get\_init\_questions}()$;
$\{a_j\} \leftarrow \text{DM.get\_answers}(\{q_j\}_{j=1}^{B^{\text{pf}}})$;
$D_0^{\text{pf}} \leftarrow \{(q_i, a_i)\}_{i=1}^{B^{\text{pf}}}$;
**for** $n = 1$ **to** $T$ **do**
  **if** $n = 1$ **then**
    **if** $P_{prior} \neq \varnothing$ **then**
      $\{x_i\}_{i=1}^{B^{\text{exp}}} \sim \text{LILO.get\_init\_x}(D_0^{\text{pf}}, P_{\text{prior}})$;
    **else**
      $\{x_i\}_{i=1}^{B^{\text{exp}}} \sim \text{Uniform}(\mathcal{X})$;
  **else**
    $\{x_i\}_{i=1}^{B^{\text{exp}}} \sim \text{acqf}(\mathcal{X}; M_{n-1})$;
  $D_n^{\text{exp}} \leftarrow D_{n-1}^{\text{exp}} \cup \{(x_i, y_i) : y_i = f(x_i)\}_{i=1}^{B^{\text{exp}}}$;
  $\{q_j\}_{j=1}^{B^{\text{pf}}} \leftarrow \text{LILO.get\_questions}(D_n^{\text{exp}}, D_{n-1}^{\text{pf}})$;
  $\{a_j\}_{j=1}^{B^{\text{pf}}} \leftarrow \text{DM.get\_answers}(\{q_j\}_{j=1}^{B^{\text{pf}}})$;
  $D_n^{\text{pf}} \leftarrow D_{n-1}^{\text{pf}} \cup \{(q_j, a_j)\}_{j=1}^{B^{\text{pf}}}$;
  $M_n \leftarrow \text{label\_data\_and\_fit\_proxy}(D_n^{\text{exp}}, D_n^{\text{pf}})$;

---

**Candidate Generation & Experimentation.** At $n = 1$, when no proxy model has been fit yet, we generate the first batch of candidates uniformly at random. For $n > 1$, using the current proxy model $M_{n-1}$, a BO acquisition function (e.g. Expected Improvement) over the search space $\mathcal{X}$ is used to select a batch of $B^{\text{exp}}$ candidate inputs $\{x_i\}_{i=1}^{B^{\text{exp}}}$. Each candidate input $x_i$ is then evaluated on the (black-box) function $f$, producing observable outcomes $y_i = f(x_i)$. After this step, the experimental dataset is updated as $D_n^{\text{exp}} = D_{n-1}^{\text{exp}} \cup \{(x_i, y_i)\}_{i=1}^{B^{\text{exp}}}$.

**Feedback Acquisition.** After obtaining new experimental outcomes, the LILO agent generates $B^{\text{pf}}$ questions for the DM to answer. These queries can include, for instance, high-level questions regarding overall optimization goals or questions about specific outcomes observed (which ones are preferred, how to improve them, etc.). The prompt for question generation contains all experimental outcomes and human feedback messages obtained. The LLM produces a set of $B^{\text{pf}}$ questions $q_j$, and the DM provides corresponding answers $a_j$. The preference feedback dataset is updated as $D_n^{\text{pf}} = D_{n-1}^{\text{pf}} \cup \{(q_j, a_j)\}_{j=1}^{B^{\text{pf}}}$.

---

**Algorithm 2:** label_data_and_fit_proxy

---

**Input:** Experimental dataset $D^{\text{exp}}$, Feedback dataset $D^{\text{pf}}$, labeling budget $K$, chunk size $S$.
$N_{\text{iter}} \leftarrow \lceil \frac{K}{S} \rceil$;
$D^{GP} \leftarrow \varnothing$;
**for** $i = 1$ **to** $N_{iter}$ **do**
  **if** $i = 1$ **then**
    $\mathcal{P} \leftarrow \text{get\_all\_pairs}(\{y : y \in D^{\text{exp}}\})$;
    $\{(y_\ell, y_\ell')\}_{\ell=1}^{S} \leftarrow \text{random\_sample}(\mathcal{P}, S)$;
  **else**
    $q \leftarrow \lceil \sqrt{2S} \rceil$;
    $\{x_i\}_{i=1}^{q} \sim \text{qEUBO}(\{x : x \in D^{\text{exp}}\}, M)$;
    $\{y_i\}_{i=1}^{q} \leftarrow \{y_j : (x_j, y_j) \in D^{\text{exp}}, x_j : \{x_i\}_{i=1}^{q}\}$;
    $\{(y_\ell, y_\ell')\}_{\ell=1}^{S} \leftarrow \text{get\_all\_pairs}(\{y_i\}_{i=1}^{q})[: S]$;
  $\mathcal{L} \leftarrow \{(y_\ell, y_\ell')\}_{\ell=1}^{S}$;
  $\{s_\ell\}_{\ell=1}^{S} \leftarrow \text{LILO.get\_pair\_labels}(\mathcal{L}, D^{\text{exp}}, D^{\text{pf}})$;
  $D^{\text{GP}} \leftarrow D^{\text{GP}} \cup \{(y_\ell, y_\ell', s_\ell)\}_{\ell=1}^{S}$;
  $M \leftarrow \text{fit\_pairwise\_GP}(D^{\text{GP}})$;
**return** $M$

---

**Utility Estimation via LLM Labeling + GP Modeling.** To convert natural language feedback from the DM into a usable optimization signal, we construct pairwise comparisons between experimental outcomes in $D^{\text{exp}}$, which are labeled with binary preference labels by the LILO agent using preferential information in $D^{\text{pf}}$. Because the number of possible outcome pairs grows quadratically with the size of $D^{\text{exp}}$, we restrict labeling to a budget of $K$ comparisons. In cases where LLM-labeling costs are a concern, we propose a sequential proxy fitting procedure that acquires labels in batches of size $S \ll K$ selected via qEUBO (Astudillo et al., 2023) (see Algorithm 2). Otherwise, for sufficiently large values of $K$, we can simply sample $K$ pairs uniformly at random. An ablation on the effect of sequential labeling with qEUBO vs random labeling is in Section 5.3.4.

The procedure runs for $N_{\text{iter}} = \lceil K/S \rceil$ iterations. At each iteration, a subset of experimental datapoints is selected using the qEUBO acquisition function under the current pairwise GP proxy model. The corresponding outcomes

are retrieved, and a batch of $S$ distinct outcome pairs are constructed from this subset. These outcome pairs are then labeled by the LILO agent to produce binary preference labels. All labeled pairs are accumulated across iterations, and after each batch the pairwise GP proxy model is refit to the full set of labeled comparisons. After the labeling budget is exhausted, the final proxy model is returned and used in the subsequent trial to guide candidate generation.

Figure 13 in the Appendix shows that the LILO agent is able to provide accurate pairwise comparisons and the accuracy continues to improve as we observe more natural language feedback. Despite this high accuracy, we do not assume noise-free labels from the LILO agent and instead leverage the pairwise GP likelihood to absorb the comparison noise.

### 4.1. Incorporating Prior Knowledge with **LILO**

In addition to providing feedback on observed outcomes, decision makers with domain expertise often possess strong prior beliefs about the optimization problem at hand. These priors may include expectations about optimal parameters $x$, or insights into how specific parameters influence outcomes $y$ – that is, information about the underlying mapping $f$. Conventional BO approaches, however, make it challenging to incorporate such priors, as this typically requires interpreting the decision maker's knowledge by a human and manually encoding it into the model and the optimization loop (e.g., through specialized kernels, custom mean functions, problem-specific acquisition functions, or carefully designed priors over the surrogate model's parameters). This process can be both time-consuming and error-prone, particularly when the expert's knowledge is qualitative and difficult to formalize mathematically. A language interface offers a more intuitive way for decision makers to express their prior knowledge.

When such prior information is available, we can warm-start LILO by replacing the uniform sampling at $n = 1$ with LLM-based sampling. Inspired by Liu et al. (2024), who show that LLMs can serve as effective candidate samplers given contextual knowledge, we generate $B^{exp}$ candidate points $x_i$ given $D_0^{pf}$ and textual prior information $P_{prior}$ via LLM prompting (see Prompt 6, Appendix B.2). Incorporating prior knowledge is optional in LILO; in Section 5.3.2, we show that a good prior can significantly improve optimization performance.

## 5. Experiments

### 5.1. Benchmarking Setup

**Test problems.** We evaluate LILO on several outcome functions $f : \mathcal{X} \to \mathcal{Y}$, subject to various utility functions $g : \mathcal{Y} \to \mathbb{R}$. In our test problems, we let $\mathcal{X} \subseteq [0, 1]^d$ and $\mathcal{Y} \subseteq \mathbb{R}^k$. Our main benchmark considers the following

synthetic and real-world outcome functions $f$:

- *DTLZ2* (Deb et al. (2002), $d = 8, k = 4$) is a synthetic outcome function commonly used as a benchmark for multi-objective optimization algorithms.
- *Vehicle Safety* (Tanabe & Ishibuchi (2020), $d = 5, k = 3$), a simulator of a vehicle's mass and two safety-defining outcomes. The inputs are the dimensions of a vehicle, which affect the vehicle's mass and safety.
- *Car Cab Design* (Liao et al. (2008), $d = 7, k = 11$), a simulator of a side-impact car crash test. The inputs measure the thickness of a car's structure, which influence the vehicle's mass, the physical impact on the passenger, and the physical impact on the car.
- *Thermal Comfort* (Fanger (1970); ISO7730 (2005), $d = 8, k = 5$) models perceptible thermal conditions and human dissatisfaction levels based on a set of environmental parameters that need to be optimized.
- *NAS-Bench-201* (Dong & Yang (2020), $d = 6, k = 4$) defines a search space of 15,625 convolutional cell architectures represented as categorical edge operations. Outcomes are four performance metrics of a given architecture (training time, accuracy, model size, latency).

We consider the following utility functions $g$ on the outcomes of these test problems:

- *piecewise linear*, modeling diminishing returns when outcomes reach their desired thresholds.
- *beta products*, describing bounded monotonic utilities that smoothly vary between increasing and decreasing marginal gains with respect to their inputs.
- *L1 distance*, measuring the L1 distance of outcomes from a predefined optimum point.
- For the Thermal Comfort problem, we consider two personas with different preferences. *Type A*: an office worker with light clothing and a moderate tolerance for different conditions; *Type B*: a summer athlete, wearing light sport kit, with a low tolerance for adverse conditions.
- For the NAS-Bench-201 problem, we consider the following two personas: *Edge*: prioritizes low latency and small model size; *Research*: prioritizes validation accuracy.

All utility functions are designed to take values in $[0, 1]$. Test problems and exact forms of the outcome and utility functions are described in detail in Appendix D.

**Baselines.** We compare LILO against two families of baselines: a) LLM-based baselines relying on the same NL feedback as LILO; b) baselines with conventional, quantitative feedback (implementation details provided in Appendix D).

**NL feedback:** ▶ **LLM (direct)**: a method relying fully on the LLM's in-context learning abilities to propose the next batch of candidates for experimentation. Given the NL feedback collected in $D_n^{pf}$ and a table of paired inputs and outputs $(x_i, y_i)$ from $D_n^{exp}$), the LLM is prompted to sug-

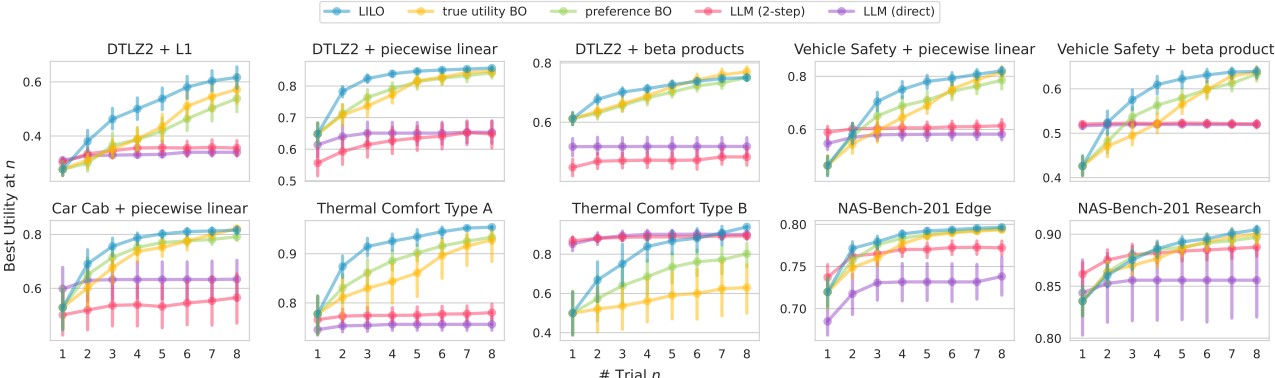

*Figure 3.* **Main benchmark.** Each panel plots the best *ground-truth* utility found so far, $\max_{x \in D_n^{\text{exp}}} g(f(x))$ over BO trial $n$ of batched experiments. LILO improves rapidly and consistently outperforms preference BO, true-utility BO, and LLM-only optimizers, especially in early trials; error bars are 95% confidence intervals over 32 replications.

gest a new batch of points $\{x_i\}_{i=1}^{B^{\text{exp}}}$. No utility-estimation step is present. ▶ **LLM (2-step)**: an ablation of LILO in which the acquisition function is replaced by LLM-based candidate generation. In each iteration, the utilities of previously observed outcomes in $D_n^{\text{exp}}$ are estimated via the LLM pairwise labeling and GP model fitting as in LILO. Given the data from $D_n^{\text{exp}}$ with the estimated utilities of outcomes and the feedback in $D_n^{\text{pf}}$, the LLM is prompted to suggest a new batch of points $\{x_i\}_{i=1}^{B^{\text{exp}}}$.

Both approaches rely on LLMs as in-context Bayesian acquisition functions as in the LLAMBO method of Liu et al. (2024). However, in LLAMBO, the ground-truth utilities of individual candidates $x_i$ are directly observable. In our setup, they must be deduced from the DM's feedback in natural language, making the setup more challenging.

**Quantitative feedback:** ▶ **true utility BO**: a GP-based BO that directly observes ground-truth utilities $u_i = g(y_i)$ for a subset of $B^{\text{pf}}$ outcomes $y_i$. ▶ **preference BO**: a setup with binary comparison feedback provided to a subset of $B^{\text{pf}}$ paired outcomes. Pairwise comparisons between two outcomes $y, y'$ are derived from the ground-truth utility differences $g(y) - g(y')$.

The subset of $B^{\text{pf}}$ feedback points / pairwise preferences is selected, as in LILO using the qEUBO acquisition function (see Appendix F.3 for an ablation on the choice of the feedback acquisition function). We also note that, in the real world, ground-truth utility values are never directly observable due to the noisy and inconsistent nature of human decision-makers. Thus, the above noiseless quantitative baselines should be seen as "gold standards".

**Benchmarking setup.** For LILO, our acquisition function of choice is the batch version of *Log Noisy Expected Improvement* (Ament et al., 2023). It is well-suited here because our proxy model is trained on noisy, LLM-derived utility estimates rather than exact evaluations of the ground-truth utility function. For fairness of comparison, for the

quantitative baselines we employ the Expected Improvement acquisition function, as the observed feedback is noise-free. Alternative options for the choice of the acquisition functions are also viable (see Appendix F.6 for a comparison with UCB and Thompson Sampling). We run the BO loop for $T = 8$ batched iterations with $B^{\text{exp}} = d$ (the dimension of the search space), simulating real-world scenarios where each evaluation of the outcome $f$ is often expensive yet parallelizable (Frazier, 2018) (see Appendix F.5 for investigation on longer-horizon experiments). For the main experiments we use a feedback batch size of $B^{\text{pf}} = 2$ (see Appendix F.1 for an ablation across varying values of $B^{\text{pf}}$). For methods involving natural language feedback, answers to questions posed by LILO are simulated with an independent LLM feedback simulator (the DM agent) that is provided with a textual description of the ground-truth utility function to ensure consistency between the NL feedback and the ground-truth utility (see Appendix D.3 for details). The simulated DM agent allows us to perform extensive and replicable experiments ablating various aspects of the proposed algorithm. In the main paper, both the DM and LILO agents are instantiated with the Llama-3.3-70b-instruct language model. Results for other choices of LLMs are presented in Appendix F.7.

### 5.2. Key Results

We first present the results across all 10 environments, where no prior knowledge about the problem is provided ($P_{\text{prior}} = \varnothing$), except for the variable names in the Vehicle Safety, Car Cab Design, and Thermal Comfort environments. For fairness of comparison with the quantitative baselines, we use LILO with random initialization, as initialization leveraging the LLM's world knowledge may give a significant performance boost as we later show in Section 5.3.2.

Figure 3 shows the maximum ground-truth utility achieved after $n$ trials of experimentation and feedback collection, that is, $\max_{x \in D_n^{\text{exp}}} g(f(x))$. We observe that LILO consistently outperforms the baselines, especially early on during

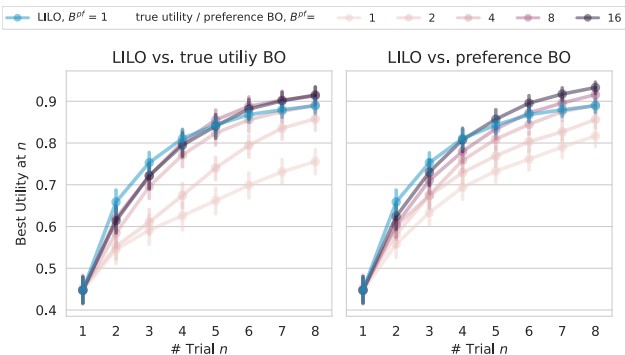

*Figure 4.* **Natural language is information-dense.** With one DM message per trial ($B^{\mathrm{pf}}=1$), LILO matches or exceeds quantitative-feedback baselines that receive $B^{\mathrm{pf}}$ labels.[1]

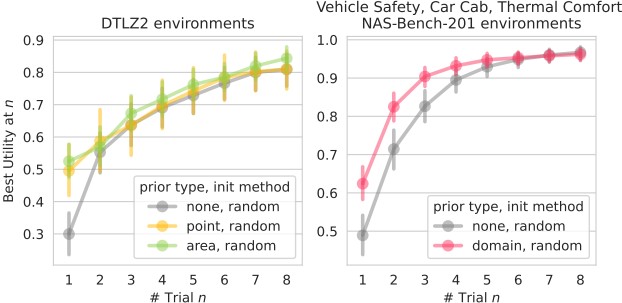

*Figure 5.* **Textual priors improve initialization.** A prior message provided only at initialization ($n=1$) warm-starts LILO: "point/area" priors specify a promising candidate or region, "domain" priors describe the semantics of a task. LLM-based initialization yields improved early-trial performance versus random.[1]

the optimization. In some environments, where outcomes correspond to semantically meaningful quantities, the two LLM baselines show good zero-shot performance, but in general, they fail to improve meaningfully over the course of the optimization. This observation aligns with the in-context learning literature on its diminishing returns as the number of examples increases (Brown et al., 2020; Zhao et al., 2024; Yin et al., 2024). Remarkably, LILO also substantially outperforms the true utility and preference-based BO baselines – this is due to the fact that **natural language feedback can convey much richer information about the overall DM's preferences than localized, point-specific feedback** as illustrated in Figure 2. The DM may provide auxiliary feedback not only on the performance of specific outcomes, but also on the overall shape of the underlying utility function (e.g., the directionality of the utility function with respect to different metrics, their relative importance, etc.). See Appendix G for example conversations from the benchmark. As the number of experimental trials increases, the advantage of natural language feedback diminishes and scalar or pairwise utility feedback baselines catch up. This effect is even more evident when running the experiments for even more iterations (see Appendix F.5).

### 5.3. Additional Studies

#### 5.3.1. THE VALUE OF NATURAL LANGUAGE FEEDBACK.

A crucial advantage of LILO is its use of natural language feedback, which can be more information-dense than conventional alternatives. It is not straightforward to compare the DM's workload necessary to answer a natural language question vs. the mental effort required to provide ratings or pairwise comparisons. The DM's burden depends on the specific questions being asked, and for conventional feedback formats it depends on the complexity of the underlying latent utility function and the dimensionality of the outcomes being judged. Experiments presented in this subsection are aimed at understanding how many pairwise comparisons or queries to the ground-truth utility are roughly equivalent to a single message of the DM in natural language. In Fig-

ure 4, we compare the performance of LILO with $B^{\mathrm{pf}} = 1$ and the quantitative baselines with $B^{\mathrm{pf}} \in \{1, 2, 4, 8, 16\}$. Results demonstrate that even one natural language statement can outperform as many as 8-16 pairwise comparisons or point-wise evaluations of the ground-truth utility. The competitive performance of LILO is most pronounced at the very initial stages of experimentation. This underscores the sample efficiency and information density of natural language feedback, when used effectively.

#### 5.3.2. INCORPORATING PRIOR KNOWLEDGE.

As described in Section 4.1, LILO can also incorporate domain priors to boost optimization performance. In the following, we demonstrate this empirically, considering three types of prior messages $P_{\mathrm{prior}}$: ▶ **point**: A message providing a sample candidate with high expected utility: *"Based on my experience, the following inputs should bring good results: $\{x\}$"*. The promising candidate is generated by sampling uniformly at random $N$ candidates, computing their ground-truth utilities, and randomly sampling a single point from the top $q\%$ of data points. ▶ **area**: A message providing expected bounds of good candidates: *"Based on my experience, inputs within these ranges should bring good results: $\{bounds\}$"*. The bounds are computed by sampling uniformly at random $N$ candidates, computing their ground-truth utilities, and taking the 0.25 and 0.75 quantiles of the top $q\%$ of data points, as the lower and upper bounds, respectively. ▶ **domain**: A message contextualizing the input parameters and the outputs. This is applicable only to semantically meaningful environments: Vehicle Safety, Car Cab Design and Thermal Comfort. Exact forms of these messages are provided in Appendix D.4. For the DTLZ2 outcome function, we apply the point and area priors with $q = 10$ and $N = 5000$. For Vehicle Safety, Car Cab Design and Thermal Comfort we apply the domain priors.

**Results.** Figure 5 presents a comparison of LILO 's performance with and without prior knowledge integrated into the optimization pipeline. As expected, **incorporating prior**

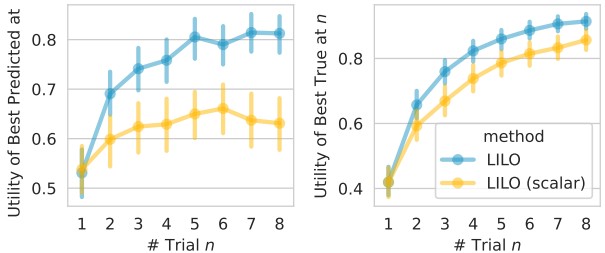

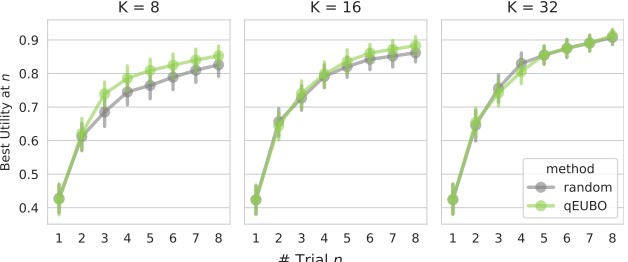

*Figure 6.* **Pairwise labeling beats direct scalar scoring.** Compared to prompting the LLM for numerical values (LILO (scalar)), LLM-based pairwise comparisons yield a more reliable proxy. Left shows the true utility of the proxy-selected best point, and right shows the best achieved utility. Pairwise supervision produces better "best-point selection" and stronger optimization progress.[1]

*Figure 7.* **Which pairs to label matters when labels are scarce.** Under a fixed budget of $K$ LLM comparisons per trial, selecting outcome pairs via qEUBO yields higher best utility when labeling budget is limited.[1]

**knowledge through LLM-based initialization substantially improves the starting point, resulting in better overall optimization performance**. For a more detailed breakdown of the results, see Appendix F.2. We also note that point and area knowledge types depend on the accuracy of externally provided information about $(g \circ f)$ and, in principle, could be incorporated into conventional BO pipelines using model-based approaches. However, LILO 's success with domain priors relies on the contextual understanding provided by the LLM's pre-training, which would be challenging to replicate with standard model-based methods. This ability to incorporate various types of prior knowledge in a unified fashion further underscores LILO 's flexibility.

### 5.3.3. PAIRWISE VS. DIRECT ESTIMATION.

The default utility estimation step in LILO relies on LLM-generated pairwise comparisons. As an alternative, we consider directly prompting the LLM to output scalar utility values. In this variant, instead of labeling pairwise preferences, the LLM produces scalar predictions $\hat{u}_i \in [0, 1]$ for each $(x_i, y_i) \in D_n^{\text{exp}}$, resulting in a dataset $D_n^{\text{GP}} = \{(x_i, \hat{u}_i)\}$. This dataset is then used to fit the proxy model $M_n$ as a standard (non-pairwise) GP. All other parts of the pipeline remain unchanged. The prompt used for this method and the modified algorithm are presented in Appendix B.

**Results.** Figure 6 reports average results over 16 replications for each environment from our benchmark. The right pane shows the maximum ground-truth utility achieved until the $n$th iteration. On the left, we report the ground-truth utility of the best candidate according to the proxy model $M_n$, i.e. $g(\hat{x}_n^*)$, where $\hat{x}_n^* = \arg\max_{x \in D_n^{\text{exp}}} M_n(x)$. In practice, it may be preferable or even necessary to use the proxy model to perform "best point selection". Especially with many observations, it may be impractical for the DM to compare a large slate of options at once – in fact, they may not be able to do this well due to mental overload. The re-

sults demonstrate that **pairwise comparisons provide more reliable utility estimates than direct scalar predictions, leading to improved optimization performance**. This observation for LLMs mirrors findings in human preference elicitation, where pairwise comparisons have been shown to yield more consistent and accurate judgments than absolute scalar ratings (Phelps et al., 2015; Hoeijmakers et al., 2024).

### 5.3.4. PAIR SELECTION WITH QEUBO

For maximal performance, experiments presented in the previous sections used a large budget ($K = 64$) to generate the LLM pairwise comparisons. In this experiment, we show LILO's performance for $K \in \{8, 16, 32\}$. We also compare our labeling method based on the qEUBO acquisition function, as presented in Algorithm 2 against the performance of a baseline with random pair selection. Across all experiments we keep the same labeling chunk size ($S = 4$).

**Results.** Figure 7 shows that as the labeling budget decreases, the advantage of qEUBO labeling increases, justifying the choice of our approach.

### 5.3.5. HIGH-DIMENSIONAL AND UNSTRUCTURED OUTCOMES.

A key feature of LILO is that it does not require an explicit surrogate model over the outcome space $\mathcal{Y}$. Instead, it only requires that an LLM can interpret realized outcomes and infer pairwise preferences over them from natural-language feedback. This makes the framework naturally compatible with *high-dimensional* and *unstructured* outcomes such as text or images. We illustrate this with an abstractive summarization task based on the CNN/DailyMail dataset (Hermann et al., 2015). In this setting, the goal is to optimize a summarization system whose controllable variables are a subset of the LLM decoding parameters (e.g., temperature, top-$p$, maximum output length), while the realized outcomes are generated textual summaries. The latent utility is defined over summary-quality indicators such as ROUGE overlap (Lin, 2004) and readability-related measures. Candidate decoding configurations are evaluated on held-out test articles that are not seen during optimization. Full details of

---

[1]Average results over all 10 environments. 32 (Figure 15) or 16 (all other figures) replications per environment, values min-max standardized within each environment before aggregation.

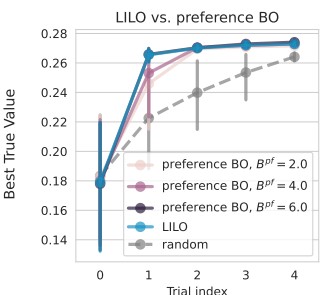
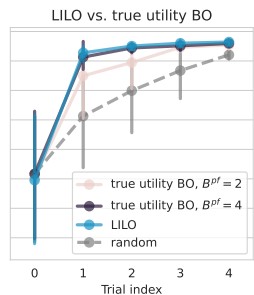

*Figure 8.* **LILO optimizes over high-dimensional and un- structured outcomes.** We evaluate LILO in a summarization benchmark where the black-box outputs are generated summaries rather than low-dimensional numerical vectors. LILO matches the strongest BO baselines that receive the maximum amount of oracle feedback, and improves more rapidly when the baselines are restricted to smaller feedback budgets.

the setup, utility function, prompts, and BO protocols are provided in Appendix F.8.

**Results.** Figure 8 reports results for the sampled-article variant, in which each BO trial uses a fresh minibatch of training articles. We compare LILO against preference BO, true-utility BO with varying feedback budgets, and random acquisition as a point of reference. In this setting the search-space dimension is $d = 4$. Following the convention $B^{exp} = d$, the maximum number of feedback data points per trial is 6 for preference BO and 4 for true-utility BO. We observe that LILO matches the strongest base- lines when those baselines are given the maximum amount of oracle feedback, and improves more rapidly when the baselines operate under smaller feedback budgets. These re- sults demonstrate that LILO can optimize successfully even when the outcomes are high-dimensional textual objects. Appendix F.8 additionally reports a fixed-article variant, in which the same training articles are reused across trials and natural-language feedback is accumulated additively across rounds; the qualitative conclusions remain the same.

### 5.3.6. ADDITIONAL ABLATION STUDIES

Appendix F contains additional results and detailed ablation studies, including analyses of feedback batch size (F.1), prior knowledge (F.2), feedback acquisition strategies (F.3), LILO's pairwise comparison accuracy (F.4), longer-horizon trials (F.5), candidate acquisition function ablations (F.6), and the use of alternative LLMs within LILO (F.7).

## 6. Discussion and Conclusion

We presented LILO, a framework for black-box optimiza- tion that enables decision makers to express goals and pref- erences in natural language rather than requiring large num- bers of pairwise comparisons or scalar ratings. Across benchmark problems, LILO achieves strong performance and we believe it is well suited to real-world optimization settings with complex multi-metric tradeoffs (e.g., online

experimentation), where explicit objectives are often un- available. LILO's performance gains arise from combining information-dense natural language feedback with proba- bilistic modeling and principled acquisition from BO. While LLM-only optimization methods can process language feedback via in-context learning, their performance often plateaus, whereas LILO sustains improvement through cali- brated uncertainty and exploration.

Several directions remain for future work. Utility estima- tion ultimately depends on the LLM's ability to interpret nuanced feedback, suggesting gains from improved prompt- ing, active question generation (Kobalczyk et al., 2025), or fine-tuning on past experimental data. While our experi- ments rely on simulated DM responses for controlled eval- uation, studies with human decision makers could provide deeper insight into real-world interaction dynamics. Hybrid approaches combining unstructured natural language with structured quantitative feedback may also mitigate limita- tions of in-context learning during utility estimation. Finally, our summarization experiment offers an initial illustration of LILO's applicability to high-dimensional unstructured outcomes such as text; a more comprehensive study of such settings, including richer text and image domains, remains an exciting direction.

## Impact Statement

This paper presents work whose goal is to advance the field of machine learning. There are many potential societal consequences of our work, none of which we feel must be specifically highlighted here.

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

# Appendix Contents

# A. Reproducibility

The code for reproducing our experimental results is available at https://github.com/facebookresearch/lilo.

# B. `LILO`: Implementation Details

### B.1. Pseudocode

Algorithm 1 presents the pseudocode of `LILO`. The `fit_proxy_models` subroutine is presented in Algorithm 2.

The algorithm for `LILO` with scalar utility estimation (experiment 5.3.3) is identical to `LILO` with pairwise preference labeling except for the `fit_proxy_models` subroutine, which is replaced with point-wise utility estimation as a scalar value in $[0, 1]$. The exact procedure is presented in Algorithm 3. All prompts used in the `LILO.xxx` subroutines are presented in Section B.2.

---

**Algorithm 1:** `LILO`: Language-in-the-Loop Optimization

---

**Require:** Max number of iterations $T$, experiment batch size $B^{\text{exp}}$, feedback batch size $B^{\text{pf}}$. **Optional:** Prior knowledge prompt $P_{\text{prior}}$

$D_0^{\text{exp}} \leftarrow \varnothing$;

$\{q_j\}_{j=1}^{B^{\text{pf}}} \leftarrow$ `LILO.get_init_questions`();

$\{a_j\} \leftarrow$ `DM.get_answers`$(\{q_j\}_{j=1}^{B^{\text{pf}}})$;

$D_0^{\text{pf}} \leftarrow \{(q_i, a_i)\}_{i=1}^{B^{\text{pf}}}$;

**for** $n = 1$ **to** $T$ **do**

    **if** $n = 1$ **then**

        **if** $P_{prior} \neq \varnothing$ **then**

            $\{x_i\}_{i=1}^{B^{\text{exp}}} \sim$ `LILO.get_init_x`$(D_0^{\text{pf}}, P_{\text{prior}})$;

        **else**

            $\{x_i\}_{i=1}^{B^{\text{exp}}} \sim \text{Uniform}(\mathcal{X})$;

    **else**

        $\{x_i\}_{i=1}^{B^{\text{exp}}} \sim \text{acqf}(\mathcal{X}; M_{n-1})$;

    $D_n^{\text{exp}} \leftarrow D_{n-1}^{\text{exp}} \cup \{(x_i, y_i) : y_i = f(x_i)\}_{i=1}^{B^{\text{exp}}}$;

    $\{q_j\}_{j=1}^{B^{\text{pf}}} \leftarrow$ `LILO.get_questions`$(D_n^{\text{exp}}, D_{n-1}^{\text{pf}})$;

    $\{a_j\}_{j=1}^{B^{\text{pf}}} \leftarrow$ `DM.get_answers`$(\{q_j\}_{j=1}^{B^{\text{pf}}})$;

    $D_n^{\text{pf}} \leftarrow D_{n-1}^{\text{pf}} \cup \{(q_j, a_j)\}_{j=1}^{B^{\text{pf}}}$;

    $M_n \leftarrow$ `label_data_and_fit_proxy`$(D_n^{\text{exp}}, D_n^{\text{pf}})$;

---

---

**Algorithm 2:** LILO `label_data_and_fit_proxy`

---

**Input:** Experimental dataset $D^{\text{exp}}$, Feedback dataset $D^{\text{pf}}$, labeling budget $K$, chunk size $S$.

$N_{\text{iter}} \leftarrow \lceil \frac{K}{S} \rceil$;

$D^{GP} \leftarrow \varnothing$;

**for** $i = 1$ **to** $N_{iter}$ **do**

  **if** $i = 1$ **then**

    $\mathcal{P} \leftarrow$ `get_all_pairs`$(\{y : y \in D^{\text{exp}}\})$;

    $\{(y_\ell, y'_\ell)\}_{\ell=1}^{S} \leftarrow$ `random_sample`$(\mathcal{P}, S)$;

  **else**

    $q \leftarrow \left\lceil \sqrt{2S} \right\rceil$;

    $\{x_i\}_{i=1}^{q} \sim$ qEUBO$(\{x : x \in D^{\text{exp}}\}, M)$;

    $\{y_i\}_{i=1}^{q} \leftarrow \{y_j : (x_j, y_j) \in D^{\text{exp}}, x_j : \{x_i\}_{i=1}^{q}\}$;

    $\{(y_\ell, y'_\ell)\}_{\ell=1}^{S} \leftarrow$ `get_all_pairs`$(\{y_i\}_{i=1}^{q})[: S]$;

  $\mathcal{L} \leftarrow \{(y_\ell, y'_\ell)\}_{\ell=1}^{S}$;

  $\{s_\ell\}_{\ell=1}^{S} \leftarrow$ LILO.`get_pair_labels`$(\mathcal{L}, D^{\text{exp}}, D^{\text{pf}})$;

  $D^{\text{GP}} \leftarrow D^{\text{GP}} \cup \{(y_\ell, y'_\ell, s_\ell)\}_{\ell=1}^{S}$;

  $M \leftarrow$ `fit_pairwise_GP`$(D^{\text{GP}})$;

**return** $M$

---

---

**Algorithm 3:** LILO (scalar) `label_data_and_fit_proxy`

---

**Input:** Experimental dataset $D^{\text{exp}}$, Feedback dataset $D^{\text{pf}}$.

$N \leftarrow |D^{\text{exp}}|$

$\{\hat{u}_i\}_{i=1}^{N} \leftarrow$ LILO.`estimate_utilities`$(\{y_i\}_{i=1}^{N}; D^{\text{exp}}, D^{\text{pf}})$

$D^{\text{GP}} \leftarrow \{(x_i, \hat{u}_i)\}_{i=1}^{N}$ $x_i$`'s are the inputs corresponding to` $y_i$`'s`

$M \leftarrow$ `fit_simple_gp`$(D^{\text{GP}})$

**return** $M$

---

## B.2. `LILO` prompts

In all prompts, `experiment_data` is a markdown-formatted table of outcomes from $D_n^{\text{exp}}$. `human_feedback` is the series of questions and answers stored in $D_n^{\text{pf}}$.

In Prompts 3 and 4, `human_feedback_summary` is a summary of the feedback stored in $D_n^{\text{pf}}$, self-generated by `LILO` using the prompt in 5. We empirically found that including this self-summarization step brings slight improvements to the resulting estimates of the LLM.

In Prompts 3 and `pair_str` is a markdown-formatted table with two rows indexed by `option_0` and `option_1`.

---

Initial question generation with `LILO.get_init_questions`

```
You are an expert in determining whether a human decision maker (DM) is going to be
satisfied with a set of experimental outcomes y = {y_names}.

## Human feedback messages:
We have also received the following messages from the DM:

{human_feedback}

## Your task:
Given the above, your task is to predict the probability of the decision maker being
satisfied with the experimental outcomes.

In order to better understand the decision maker's utility function you want to ask
them about their optimization goals.

Provide a list of questions you would ask the decision maker to better understand their
 internal utility model.

Return your final answer a a json file with the following format containing exactly {
n_questions} most important questions:
```json
{{
    "q1" : <question1>,
    ...
    "q{n_questions}" : <question{n_questions}>
}}
```
```

---

*Prompt 1.* The prompt used for question generation in the `LILO.get_init_questions` subroutine.

---

Question generation with `LILO.get_questions`

```
You are an expert in determining whether a human decision maker (DM) is going to be
satisfied with a set of experimental outcomes y = {y_names}.

## Experimental outcomes:
So far, we have obtained the following experimental outcomes:

{experiment_data}

## Human feedback messages:
We have also received the following messages from the DM:

{human_feedback}

## Your task:
Given the above, your task is to predict pairwise preferences between experimental
outcomes.

In order to better understand the decision maker's utility function you want to ask
them about their optimization goals or for feedback regarding specific experimental
outcomes.

First, analyze the decision maker's goals and feedback messages to understand their
overall preferences.
Then, provide a list of questions you would ask the decision maker to better
understand their internal utility model.
Your questions can be either general or referring to specific outcomes. For instance,
you may ask the decision maker:
- questions clarifying the optimization objective,
- to rank two (or more) outcomes,
- how to improve certain outcomes,
- for a Likert-scale rating regarding a specific outcome,
- etc.
When referring to specific outcomes, always state the arm_index involved.
Your questions should help you predict pairwise preferences between any two
experimental outcomes from the set of experimental outcomes provided above.

Return your final answer a a json file with the following format containing exactly {
n_questions} most important questions:
```json
{{
    "q1" : <question1>,
    ...
    "q{n_questions}" : <question{n_questions}>
}}
```

*Prompt 2.* The prompt used for question generation in the `LILO.get_questions` subroutine.

---

---

Pairwise comparisons with `LILO.get_pair_labels`

```
You are an expert in determining whether a human decision maker (DM) is going to be
satisfied with a set of experimental outcomes y = {y_names}.

## All experimental outcomes:

{experiment_data}

## Human feedback messages:
We have also received the following messages from the DM:

{human_feedback}

{human_feedback_summary}

## Your task:
Given a pair of outcomes--option_0 and option_1, your goal is to decide which one is
more preferable according to the DM's preferences.

{pair_str}

Provide your prediction as a json file with the following format:
```json
{{
    "reasoning": "Your reasoning about the DM's preferences and option_0 vs. option_1.
 Do not insert new lines in your reasoning.",
    "answer" : 0 or 1
}}
```
where in "answer" you should return 0 if option_0 is preferred, or 1 if option_1 is
preferred.
Return just the json file (with the header ```json), nothing else.
```

*Prompt 3.* The prompt used for pairwise comparison labeling used in the `LILO.get_pairwise_pref` subroutine.

Scalar utility estimation with with `LILO.estimate_utilities`

```
You are an expert in determining whether a human decision maker (DM) is going to be
satisfied with a set of experimental outcomes y = {y_names}.

## Experimental outcomes:
So far, we have obtained the following experimental outcomes:

{experiment_data}

## Human feedback messages:
We have also received the following messages from the DM:

{human_feedback}

{human_feedback_summary}

## Your task:
Given the above, your task is to predict the probability of the decision maker being
satisfied with the experimental outcomes.

First, analyse the human feedback messages to understand the DM's preferences.
Then, provide your predictions for all y's in the set of all experimental outcomes
above.
Return your final answer as a jsonl file with the following format:

'''jsonl
{{
    "arm_index": "{idx0}",
    "reasoning": <reasoning>,
    "p_accept": <probability>
}}
{{
    "arm_index": "{idx1}",
    "reasoning": <reasoning>,
    "p_accept": <probability>
}}
...
{{
    "arm_index": "{idxn}",
    "reasoning": <reasoning>,
    "p_accept": <probability>
}}
'''
Where <reasoning> should be a short reasoning for your prediction and <probability>
should be your best estimate for the probability between 0 and 1 that the DM will be
satisfied with the corresponding outcome.

Provide your predictions for ALL y's in the set of experimental outcomes above. That
is, for EACH outcome from {idx0}. to {idxn}.
Do not generate any Python code. Just return your predictions as plain text.
```

*Prompt 4.* The prompt used for scalar utility estimation used in the `LILO.estimate_utilities` subroutine.

---

Human Feedback Summarization

```
You are an expert in determining whether a human decision maker (DM) is going to be
satisfied with a set of experimental outcomes y = {y_names}.

## Experimental outcomes:
So far, we have obtained the following experimental outcomes:

{experiment_data}

## Human feedback messages:
We have also received the following messages from the DM:

{human_feedback}

## Your task:
Given the above, your task is to summarize the human feedback messages into a clear
description of the DM's optimization goals.
Make your summary as quantitative as possible so that it can be easily used for utility
 estimation.

After analyzing the human feedback messages, return your final answer as a json file
with the following format:
'''json
{{
    "summary": <summary>
}}
'''
Remember about the '''json header!
```

*Prompt 5.* The prompt used for generating the human_feedback_summary by LILO for pairwise comparisons or scalar utility estimation.

Candidate Generation at $n = 1$ when $P_{\text{prior}} \neq \varnothing$, `LILO.get_init_x`

```
You are performing optimization of a utility function u(x) = g(y) = g(f(x)), where x
is a vector of parameters: x = {x_names} and y = f(x) = {y_names} is a vector of
outcomes.
Each dimension of x is in the range [0, 1].
Your goal is to find the parameters x that maximize the utility.

## Prior knowledge:
You have obtained the following prior knowledge about the experiment:
{prior_knowledge}

## Human feedback messages:
You have also received the following messages from the DM:
{human_feedback}

## Your task:
Given the above, your task is to generate a set of {n_candidates} candidate parameters
 x for the next round of experimentation.

First, analyse the information above, then return your final answer as a json file
with the following format:
'''json
{{
    "0": <candidate0>,
    "1": <candidate1>,
    ...
    "{n}": <candidate{n}>,
}}
'''
Where each <candidatei> is a list of the candidate parameter values in [0, 1].
Do not write Python code for candidate generation. Just return the required json.
Do not add any comments to your json. Remember about the '''json header.
```

*Prompt 6.* The prompt used for candidate generation by `LILO` at $n = 1$ when prior knowledge is available (`LILO.sample_init_candidates` subroutine).

### B.3. GP models and acquisition functions

In our implementation of the algorithm we rely on the BoTorch Python library (Balandat et al., 2020) to implement the subroutines of GP model fitting, acquisition function evaluation and candidate generation. Specifically, proxy GP models are instances of `PairwiseGP` or `SingleTaskGP` classes, with their default settings (with the exception for the NAS-Bench-201 environment, where the default GP kernels are replaced with categorical kernels due to the environment structure).

Main experiments are presented for the LogNEI acquisition function for candidate generation and qEUBO acquisition function for feedback datapoint selection. Both functions are used in their batched-versions, which jointly evaluate the utility of an entire batch of candidates via Monte Carlo sampling rather than selecting points greedily. Specifically, we employ the `qLogNoisyExpectedImprovement` method from BoTorch, which follows the work of Ament et al. (2023) and the `qExpectedUtilityOfBestOption` (Lin et al., 2022).

## C. Baselines: Implementation Details

### C.1. Natural Language Feedback Baselines

We implement two versions of an LLM-based approach to candidate generation for optimization.

**LLM (2-step)** follows the same Algorithm 1 as `LILO` with the following exceptions. In the candidate generation step, instead of using the LogNEI acquisition function, we prompt the LLM to generate a set of candidates using prompt 7.

**LLM (direct)** omits the step of utility estimation and generates the candidates directly based on the raw human feedback and observed inputs and outcomes from $D_n^{exp}$. The prompt used is presented in Prompt 8

---

Candidate Generation, LLM (2-step)

```
You are performing optimization of a utility function u(x) = g(y) = g(f(x)), where x
is a vector of parameters: x = {x_names} and y = f(x) = {y_names} is a vector of
outcomes.
Each dimension of x is in the range [0, 1].
Your goal is to find the parameters x that maximize the utility.

## Experimental Outcomes
So far, you have also observed the following inputs x and their estimated utilities:

{experiment_data}

## Human feedback messages:
We have also received the following messages from the DM:

{human_feedback}

## Your task:
Given the above, your task is to generate a set of {n_candidates} candidate parameters
 x for the next round of experimentation.
Your candidates should maximize the expected improvement over the current best
candidate x^* = {x_star} with utility u(x^*) = {u_star}.

First, analyse the information above, then return your final answer as a json file
with the following format:
```json
{{
    "0": <candidate0>,
    "1": <candidate1>,
    ...
    "{n}": <candidate{n}>,
}}
```
Where each <candidatei> is a list of the candidate parameter values in [0, 1].
Do not write Python code for candidate generation. Just return the required json.
Do not add any comments to your json. Remember about the ```json header.
```

---

*Prompt 7.* The prompt used for candidate generation by the LLM (2-step) baseline. In the above, `experiment_data` is a markdown formatted table of outcomes $y_i$ and their estimate utilities via the LLM-based proxy model $M(x_i)$. `x_star` and `u_star` are determined based on the estimated utilities (not ground-truth $g(y_i)$'s as these are latent, non-observable quantities).

---

Candidate Generation, LLM (direct)

```
You are performing optimization of a utility function u(x) = g(y) = g(f(x)), where x
is a vector of parameters: x = {x_names} and y = f(x) = {y_names} is a vector of
outcomes.
Each dimension of x is in the range [0, 1].
Your goal is to find the parameters x that maximize the utility.

{experiment_data}

## Human feedback messages:
We have also received the following messages from the DM:

{human_feedback}

## Your task:
Given the above, your task is to generate a set of {n_candidates} candidate parameters
 x for the next round of experimentation.
First, analyze the human feedback messages to understand the DM's preferences.
Then, generate a set of {n_candidates} candidate parameters x, trading-off exploration
 and exploitation.
Return your final answer as a json file with the following format:
```json
{{
    "0": <candidate0>,
    "1": <candidate1>,
    ...
    "{n}": <candidate{n}>,
}}
```
Where each <candidatei> is a list of the candidate parameter values: {x_names}, each
in [0, 1].
Do not write Python code for candidate generation. Just return the required json.
Do not add any comments to your json.
```

*Prompt 8.* The prompt used for candidate generation in the LLM (direct) baseline. experiment_data is a markdown-formatted table of inputs and outcomes. human_feedback is the set of questions and answers from $D_{n-1}^{\text{pf}}$.

### C.2. Quantitative Feedback Baselines

The quantitative baseline methods follow an analogous procedure to LILO, where the Q&A natural language feedback is replaced with either scalar values of the utilities associated with a batch of outcomes (true utility BO) or pairwise comparisons between outcomes based on their ground-truth utilities (preference BO). This feedback is obtained on $B^{\text{pf}}$ outcomes $y_i$ or paired outcomes $(y_i, y'_i)$ sampled with the qEUBO acquisition function (see Appendix F.3 for a justification of this choice). In LILO, the LLM extends the natural language feedback to the observations in $D_n^{\text{exp}}$ via pairwise labeling. For our quantitative baselines, to extend the quantitative feedback within $D^{\text{pf}}$ to the entirety of $D_n^{\text{exp}}$, we fit a simple / pairwise GP model $M_n^y : \mathcal{Y} \to \mathcal{P}(\mathbb{R})$. Subsequently a proxy model $M_n^x : \mathcal{X} \to \mathcal{P}(\mathbb{R})$ is fit to the predictions of the $M_n^y$ model.

We present the exact implementation of these methods in Algorithms 4 and 5.

---

**Algorithm 4:** True utility BO

---

**Input:** Max number of iterations $T$, Experiment batch size $B^{\text{exp}}$, Feedback batch size $B^{\text{pf}}$.
$D_0^{\text{exp}} \leftarrow \varnothing$
$D_0^{\text{pf}} \leftarrow \varnothing$
**for** $n = 1$ **to** $T$ **do**
   # Sample a batch of candidates
   **if** $n = 1$ **then**
      $\lfloor \ \{x_i\}_{i=1}^{B^{\text{exp}}} \sim \text{Uniform}(\mathcal{X})$
   **else**
      $\lfloor \ \{x_i\}_{i=1}^{B^{\text{exp}}} \sim \text{LogEI}(\mathcal{X}; M_{n-1}^x)$
   # Run the experiments and update the experimental dataset
   $D_n^{\text{exp}} \leftarrow D_{n-1}^{\text{exp}} \cup \{(x_i, y_i) : y_i = f(x_i)\}_{i=1}^{B^{\text{exp}}}$
   # Sample a batch of outcomes for feedback
   **if** $n = 1$ **then**
      $\lfloor \ \{y_i\}_{i=1}^{B^{\text{pf}}} \sim \texttt{random\_sample}(D^{\text{exp}})$
   **else**
      $\lfloor \ \{y_i\}_{i=1}^{B^{\text{pf}}} \sim \text{qEUBO}(D_n^{\text{exp}}, M_{n-1}^y)$
   # Update the preference feedback dataset
   $D_n^{\text{pf}} \leftarrow D_{n-1}^{\text{pf}} \cup \{(y_j, u_j) : u_j = g(y_j)\}_{j=1}^{B^{\text{pf}}}$
   # Fit a Y->U GP
   $M_n^y \leftarrow \texttt{fit\_simple\_gp}(D_n^{\text{pf}})$
   # Label all experimental datapoints
   $D_n^{\text{GP}} \leftarrow \{(x_i, y_i, \hat{u}_i) : (x_i, y_i) \in D_n^{\text{exp}}\}$ // $\hat{u}_i$'s are the mean predictions with respect to
      $M_n^y : \mathcal{Y} \to \mathcal{P}(\mathbb{R})$
   # Fit the proxy model
   $D_n^{\text{GP},x} \leftarrow \{(x_i, \hat{u}_i) : (x_i, y_i, \hat{u}_i) \in D_n^{\text{GP}}\}$
   $M_n^x \leftarrow \texttt{fit\_simple\_gp}(D_n^{GP,x})$

---

---

**Algorithm 5:** Preference BO

---

**Input:** Max number of iterations $T$, Experiment batch size $B^{\text{exp}}$, Feedback batch size $B^{\text{pf}}$.

$D_0^{\text{exp}} \leftarrow \varnothing$,

$D_0^{\text{pf}} \leftarrow \varnothing$

**for** $n = 1$ *to* $T$ **do**

    # Sample a batch of candidates

    **if** $n = 1$ **then**

        $\{x_i\}_{i=1}^{B^{\text{exp}}} \sim \text{Uniform}(\mathcal{X})$

    **else**

        $\{x_i\}_{i=1}^{B^{\text{exp}}} \sim \text{LogNEI}(\mathcal{X}; M_{n-1}^x)$

    # Run the experiments and update the experimental dataset

    $D_n^{\text{exp}} \leftarrow D_{n-1}^{\text{exp}} \cup \{(x_i, y_i) : y_i = f(x_i)\}_{i=1}^{B^{\text{exp}}}$

    # Sample a batch of paired outcomes for feedback

    **if** $n = 1$ **then**

        $\{(y_i, y_i')\}_{i=1}^{B^{\text{pf}}} \sim \texttt{random\_sample}(D^{\text{exp}})$

    **else**

        $\{(y_i, y_i')\}_{i=1}^{B^{\text{pf}}} \sim \text{qEUBO}(D_n^{\text{exp}}, M_{n-1}^y)$

    # Update the preference feedback dataset

    $D_n^{\text{pf}} \leftarrow D_{n-1}^{\text{pf}} \cup \{(y_j, y_j', p_j) : p_j = \mathbb{1}\{g(y_j) > g(y_j')\}\}_{j=1}^{B^{\text{pf}}}$

    # Fit a Y->U GP

    $M_n^y \leftarrow \texttt{fit\_pairwise\_gp}(D_n^{\text{pf}})$

    # Label all experimental datapoints

    $D_n^{\text{GP}} \leftarrow \{(x_i, y_i, \hat{u}_i) : (x_i, y_i) \in D_n^{\text{exp}}\}$ // $\hat{u}_i$'s are the mean predictions with respect to

        $M_n^y : \mathcal{Y} \rightarrow \mathcal{P}(\mathbb{R})$

    # Fit the proxy model

    $D_n^{\text{GP},x} \leftarrow \{(x_i, \hat{u}_i) : (x_i, y_i, \hat{u}_i) \in D_n^{\text{GP}}\}$

    $M_n^x \leftarrow \texttt{fit\_simple\_gp}(D_n^{GP,x})$

---

# D. Simulation Environments

We benchmark LILO on synthetic and real-world outcome functions as well as several utility functions. In all our test problems we have $\mathcal{X} = [0,1]^d$ and $\mathcal{Y} = \mathbb{R}^k$. The main benchmark considers in total 10 environments, represented as outcome–utility function pairs. All outcome and utility functions are described in detail in this section.

In our simulations, we run the BO loop for $T$ iterations, setting $B^{\mathrm{exp}} = d$ (the dimension of the search space). We seed the conversation between the DM and LILO by hard-coding the first message of the DM agent. Details of each outcome and utility functions alongside the seeding messages are presented below.

## D.1. Outcome Functions

**DTLZ2**  The DTLZ2 function was introduced by (Deb et al., 2002), allowing for arbitrary input dimension $d$ and output dimension $k$ subject to $d > k$. $\mathcal{X} = [0,1]^d$. For a DTLZ2 function $f$ with a $k$-dimensional output and $d$-dimensional input, we have:

$$m := d - k + 1$$

$$h(x) := \sum_{i=m}^{d-1} (x_i - 0.5)^2$$

$$f_j(x) = -(1 + h(x)) \left( \prod_{i=1}^{k-j-1} \cos\left(\frac{\pi}{2}x_i\right) \right) \mathbb{1}_{j>1} \sin\left(\frac{\pi}{2}x_{k-j-1}\right)$$

In our experiments we use $d = 8$ and $k = 4$.

**Vehicle Safety**  This is a test problem for optimizing vehicle crash-worthiness with $d = 5$ and $k = 3$. $\mathcal{X} = [1,3]^5$. We refer readers to Tanabe & Ishibuchi (2020) for details on function definition. We normalize each component of $y = f(x)$ to lie between 0 and 1 based on empirical bounds on the outcome space $\mathcal{Y}$.

**Car Cab Design**  We refer readers to Liao et al. (2008) for details. Note that in the original problem, there are stochastic components which we replace with deterministic components fixed at their original mean values in order to obtain a deterministic ground-truth outcome function. We normalize each component of $y = f(x)$ to lie between 0 and 1 based on empirical bounds on the outcome space $\mathcal{Y}$.

**Thermal Comfort**  The problem setting follows the ISO 7730 and ASHRAE 55 models, which predict human thermal sensation and dissatisfaction based on six core factors: air temperature, mean radiant temperature, humidity, air speed, clothing insulation, and metabolic rate (Fanger, 1970; ISO7730, 2005; ASHRAE55, 2020). From these, outcome measures such as Predicted Percentage Dissatisfied (PPD), draft risk (DR), vertical air temperature difference, radiant temperature asymmetry, and floor surface temperature are derived, each with threshold values associated with acceptable comfort. The goal of the optimization agent is to find environmental parameters that minimize discomfort and keep all outcomes within desirable ranges, reflecting realistic expectations of the occupant rather than arbitrary synthetic functions. This setting is widely used in thermal comfort research and can be visualized via the CBE Thermal Comfort Tool (Tartarini et al., 2020). In our implementation, the outcome function has two fixed, non-optimizable parameters: clothing insulation (clo $\in [0.3, 1.2]$) and metabolic rate (met $\in [1.0, 2.0]$) which differ for the two versions of the environments considered in this paper, as detailed in the next section.

**NAS-Bench-201**  The problem setting is based on NAS-Bench-201 (Dong & Yang, 2020), a tabular neural architecture search benchmark containing a fixed search space of 15,625 convolutional cell architectures together with pre-computed performance statistics for each architecture. Each architecture is represented by a six-dimensional categorical vector specifying the operation assigned to each edge of a directed acyclic graph cell, and is associated with four outcome measures: training time, validation accuracy, model size, and inference latency. These outcomes induce realistic trade-offs between predictive performance and deployability: larger and slower models often achieve higher accuracy, whereas smaller and faster models are preferable for deployment on constrained devices. The goal of the optimization agent is therefore to identify architectures that best match the latent preferences of a decision maker over these competing objectives, rather than simply maximizing accuracy in isolation. This benchmark is particularly attractive for our setting because it combines

a purely categorical search space with nonlinear, persona-dependent utility functions over multiple realistic performance metrics, while remaining fully reproducible through tabular evaluation. Because the search space is purely categorical, all GPs applied to this environment (both in `LILO` and baselines) use categorical kernels based on the Hamming distance.

### D.2. Utility Functions

**L1 distance**   We consider a utility function which is the L1 distance from a pre-specified point $y_{\text{opt}}$. This choice of the utility function mimics the scenario where the DM wishes to keep the outcomes close to a specific desirable state.

For DTLZ2, we set $y_{\text{opt}} = [0.8, 1.0, 0.7, 1.25]$.

The message seeding the conversation takes the following form:

> **Goal message (L1 distance)**
>
> ```
> My goal is to bring all the outcome metrics as close to {opt_y} as possible.
> ```

**Beta products**   Prior work on preference learning has utilized the Beta CDF to form utility functions. The Beta CDF provides a convenient, bounded monotonic transform that smoothly varies between increasing and decreasing marginal gains with respect to their inputs. Our utility function takes the following form:

$$g(y; \alpha, \beta) = \prod_{i=1}^{k} \text{BetaCDF}(y_i; \alpha_i, \beta_i)$$

For the DTLZ2 outcome function we set:

$$\alpha = [0.5, 2.0, 2.0, 2.0]$$
$$\beta = [0.5, 1.0, 2.0, 5.0]$$

For the Vehicle Safety outcome function we set:

$$\alpha = [0.5, 1.0, 1.5]$$
$$\beta = [1, 2, 3]$$

For this utility function, the message seeding the conversation takes the following form:

> **Goal message (beta products)**
>
> ```
> My goal is to bring all the outcome metrics as close to 1 as possible. Results are
> strongest only when every metric is high -- if any metric is low, it significantly
> reduces the overall performance.
> ```

**Piecewise linear**   Analogously to (Lin et al., 2022) we also consider a piecewise linear utility function. Its shape corresponds to diminishing marginal returns on outcomes and sharp declines in utility when desired thresholds are not met. For a $k$-dimensional input vector $y$, this utility function is defined as:

$$g(y; \beta_1, \beta_2, t) = \sum_{i=1}^{k} h_i(y_i),$$

where

$$h_i(y_i) = \begin{cases} \beta_{1,i} y_i + (\beta_{1,i} - \beta_{2,i}) t_i & \text{if } y_i < t_i \\ \beta_{2,i} y_i & \text{if } y_i \geq t_i \end{cases}.$$

For the DTLZ2 problem, we set

$$\beta_1 = [4.0, 3.0, 2.0, 1.0]$$
$$\beta_2 = [0.4, 0.3, 0.2, 0.1]$$
$$t = [1.0, 0.8, 0.5, 0.5]$$

For the Vehicle Safety problem, we set

$$\beta_1 = [2, 6, 8]$$
$$\beta_2 = [1, 2, 2]$$
$$t = [0.5, 0.8, 0.8]$$

For the Car Cab Design problem, we set

$$\beta_1 = [7.0, 6.75, 6.5, 6.25, 6.0, 5.75, 5.5, 5.25, 5.0, 4.75, 4.5]$$
$$\beta_2 = [0.5, 0.4, 0.375, 0.35, 0.325, 0.3, 0.275, 0.25, 0.225, 0.2, 0.175]$$
$$t = [0.64, 0.68, 0.96, 0.88, 1.06, 0.65, 0.84, 0.86, 0.58, 0.7, 0.53]$$

Here, thresholds $t_i$ correspond to the 0.75 quantiles of the outcome values $y_i$. The seeding message takes the following form:

> **Goal message (piecewise linear)**
>
> ```
> My goal is to achieve the following thresholds in each outcome {t}. Improvements over
> the thresholds are always good, but less important than bringing the outcomes to their
>  threshold values. The further away an outcome is from its threshold, the higher is
> its negative impact on the overall performance.
> ```

**Thermal Comfort** Our utility maps the outcome vector $Y = [\text{PPD}, \text{DR}, dT_{\text{vert}}, dT_{\text{pr}}, T_{\text{floor}}]$ to a scalar $U \in [0, 1]$ via per–outcome desirabilities that enforce being "within range", using the Derringer-Suich desirability functions. For the four "smaller is better" outcomes (PPD, DR, $dT_{\text{vert}}$, $dT_{\text{pr}}$) we use a one–sided acceptable band with a comfort threshold $L$ and an unacceptable threshold $H$ and define

$$d_{\text{small}}(y; L, H, s) = \begin{cases} 1, & y \leq L, \\ \left( \dfrac{H - y}{H - L} \right)^s, & L < y < H, \\ 0, & y \geq H, \end{cases}$$

so values at or below $L$ are fully desirable, values beyond $H$ are unacceptable, and values in between taper smoothly with shape $s \geq 1$. For floor temperature $T_{\text{floor}}$ we target a comfort band $[l, h]$ and tolerate a wider band $[l_{\min}, h_{\max}]$ by

$$d_{\text{band}}(t; l, h, l_{\min}, h_{\max}, s) = \begin{cases} 1, & l \leq t \leq h, \\ \left( \dfrac{t - l_{\min}}{l - l_{\min}} \right)^s, & l_{\min} < t < l, \\ \left( \dfrac{h_{\max} - t}{h_{\max} - h} \right)^s, & h < t < h_{\max}, \\ 0, & t \leq l_{\min} \text{ or } t \geq h_{\max}. \end{cases}$$

The overall utility is the geometric mean of the five desirabilities,

$$U(Y) = \Big( d_{\text{small}}(\text{PPD}) \cdot d_{\text{small}}(\text{DR}) \cdot d_{\text{small}}(dT_{\text{vert}}) \cdot d_{\text{small}}(dT_{\text{pr}}) \cdot d_{\text{band}}(T_{\text{floor}}) \Big)^{1/5},$$

.

We consider two versions of this utility function with varying comfortable ranges of the outcome metrics

**Type A**. These settings are meant to simulate preferences of an office worker in light clothing and a moderate tolerance for different conditions.

$$l_{\text{PPD}} = 0.0, \; h_{\text{PPD}} = 30.0,$$
$$l_{\text{DR}} = 10.0, \; h_{\text{DR}} = 35.0,$$
$$l_{\text{dT}_{\text{vert}}} = 3.0, \; h_{\text{dT}_{\text{vert}}} = 9.0,$$
$$l_{\text{dT}_{\text{pr}}} = 5.0, \; h_{\text{dT}_{\text{pr}}} = 22.0,$$
$$l_{\text{min},\text{T}_{\text{floor}}} = 16.0, \; l_{\text{T}_{\text{floor}}} = 19.0, \; h_{\text{T}_{\text{floor}}} = 26.0, \; h_{\text{min},\text{T}_{\text{floor}}} = 30.0.$$

In Thermal Comfort Type A the clothing and metabolic rate parameters of the outcome function are set to $\text{clo} = 0.61$ and $\text{met} = 1.0$, respectively.

**Type B**. These settings are meant to simulate preferences of a summer athlete wearing light sport kit, with a lower tolerance for adverse conditions.

$$l_{\text{PPD}} = 0.0, \; h_{\text{PPD}} = 24.0,$$
$$l_{\text{DR}} = 30.0, \; h_{\text{DR}} = 45.0,$$
$$l_{\text{dT}_{\text{vert}}} = 2.5, \; h_{\text{dT}_{\text{vert}}} = 6.0,$$
$$l_{\text{dT}_{\text{pr}}} = 4.0, \; h_{\text{dT}_{\text{pr}}} = 12.0,$$
$$l_{\text{min},\text{T}_{\text{floor}}} = 19.0, \; l_{\text{T}_{\text{floor}}} = 20.0, \; h_{\text{T}_{\text{floor}}} = 23.0, \; h_{\text{min},\text{T}_{\text{floor}}} = 25.0.$$

In Thermal Comfort Type B the clothing and metabolic rate parameters of the outcome function are set to $\text{clo} = 0.3$ and $\text{met} = 2.0$, respectively.

The seeding message takes the following form:

---

Goal message (Thermal Comfort)

```
My goal is to keep all metrics within my thermal comfort preferences.
```

---

**NAS-Bench-201** Our utility maps the outcome vector

$$Y = [y_0, y_1, y_2, y_3] = [\text{train\_time}, \; \text{val\_acc}, \; \text{model\_size}, \; \text{latency}]$$

to a scalar $U \in [0, 1]$ using smooth per-metric contributions, an interaction term between accuracy and latency, and penalties for architectures that are excessively large or slow. Each outcome is first transformed through a sigmoid-based desirability function centered around a preference threshold $\tau_i$ with softness parameter $\beta_i$:

$$c_i(y_i; \tau_i, \beta_i) = \sigma\left(\frac{y_i - \tau_i}{\beta_i}\right) - \frac{1}{2},$$

where $\sigma(\cdot)$ is the logistic sigmoid. Since lower values are preferable for training time, model size, and latency, their contributions are sign-reversed. The latent score is then

$$U_{\text{latent}}(Y) = \sum_{i=0}^{3} w_i c_i(Y) + \alpha \, c_{\text{acc}}(y_1) c_{\text{lat}}(y_3) - \lambda_s \max(0, y_2 - s_{\text{max}})^2 - \lambda_l \max(0, y_3 - l_{\text{max}})^2,$$

where $w_i$ are metric weights, $\alpha$ controls the interaction between accuracy and latency, and the last two terms penalize model sizes and latencies that exceed the maximum acceptable values $s_{\text{max}}$ and $l_{\text{max}}$. Finally, the utility is mapped to $[0, 1]$ via

$$U(Y) = \sigma\left(\frac{U_{\text{latent}}(Y) - b}{s}\right),$$

where $b$ and $s$ are bias and scale parameters. This formulation captures smooth preferences, diminishing returns, and soft but increasingly severe penalties beyond practical deployment limits.

We consider two versions of this utility function corresponding to two different decision-maker profiles.

**Edge Persona.** These settings are meant to simulate preferences of a user deploying models on a resource-constrained edge device, where low latency and small model size are critical and validation accuracy matters primarily among architectures that are already deployable:

$$(w_0, w_1, w_2, w_3) = (0.08, \ 0.22, \ 0.30, \ 0.40),$$
$$(\tau_0, \tau_1, \tau_2, \tau_3) = (0.42, \ 0.94, \ 0.16, \ 0.43),$$
$$(\beta_0, \beta_1, \beta_2, \beta_3) = (0.10, \ 0.035, \ 0.06, \ 0.05),$$
$$\alpha = 0.22, \qquad \lambda_s = 7.0, \qquad \lambda_l = 8.0,$$
$$s_{\max} = 0.28, \qquad l_{\max} = 0.52, \qquad b = -0.20, \qquad s = 0.50.$$

This utility strongly rewards architectures that are fast and compact, and imposes steep penalties when latency or model size exceed deployability limits.

The seeding messages take the following form:

---

Goal message (NAS-Bench-201 Edge)

```
My goal is to find an architecture that is practical for deployment on a resource-
constrained edge device. Low latency and small model size matter most to me. Accuracy
is still important, but only after the model is fast enough and compact enough to be
deployable. I also prefer shorter training times, although that is usually less
important than deployment constraints.
```

---

**Research Persona.** These settings are meant to simulate preferences of a researcher primarily interested in predictive performance, while still preferring architectures that are reasonably efficient:

$$(w_0, w_1, w_2, w_3) = (0.20, \ 0.60, \ 0.10, \ 0.10),$$
$$(\tau_0, \tau_1, \tau_2, \tau_3) = (0.46, \ 0.96, \ 0.30, \ 0.55),$$
$$(\beta_0, \beta_1, \beta_2, \beta_3) = (0.10, \ 0.02, \ 0.08, \ 0.08),$$
$$\alpha = 0.10, \qquad \lambda_s = 2.5, \qquad \lambda_l = 3.0,$$
$$s_{\max} = 0.60, \qquad l_{\max} = 0.70, \qquad b = 0.00, \qquad s = 0.20.$$

Compared to the edge persona, this utility places much more emphasis on validation accuracy and uses softer constraints on model size and latency.

The seeding messages take the following form:

---

Goal message (NAS-Bench-201 Research)

```
My goal is to find an architecture with the strongest predictive performance.
Validation accuracy is the main thing I care about. I do also care about training time
 because faster iteration is useful, and I would rather avoid unnecessarily large or
slow models, but those concerns are secondary unless they become excessive.
```

---

### D.3. LLM-based simulation of the human preference feedback

Evaluating black box optimization algorithm rigorously is challenging due to the replications required to discern performance in the presence of the inherent variance of the evaluation of the optimization traces. This is exacerbated in our setting where the algorithm is based on feedback from human decision makers. To be able to evaluate `LILO` rigorously, we therefore simulate the human preference feedback with another LLM.

For all our experiments we use Llama-3.3-70b-instruct as the language model for the human feedback simulator. A high-level representation of the prompt used to generate the answers is presented in Prompt 9. `utility_func_desc` is a textual description of the specific utility function. `outcomes_markdown` is a markdown-formatted table of outcomes $y_i \in D^{\text{exp}}$ and their corresponding pre-computed ground-truth utilities $g(y_i)$. `questions_str` are the questions $\{q_j\}_{j=1}^{B^{\text{pf}}}$ generated by `LILO`. Finally `utility_constraints` contain additional utility-specific instructions for the LLM to generate human-like feedback and to not reveal explicitly the exact functional form of the utility, ensuring the generated answers sound natural.

### D.4. Prior Knowledge Messages $P_{\text{prior}}$

Below we present the prior messages $P_{\text{prior}}$ used in our experiments from Section 5.3.2.

---

DTLZ2 (point knowledge)

```
- Based on my experience, the following inputs should bring good results: {
promising_point}.
```

---

DTLZ2 (area knowledge)

```
- Based on my experience, inputs within these ranges should bring good results {bounds
}:
```

---

Vehicle Safety (domain knowledge)

```
- y_1 measures the reduction in vehicle's mass, y_2 measures the reduction in
integration of acceleration between two time points, y_3 measures the reduction in toe
 board intrusion in the offset-frontal crash.
- The parameters x measure the thickness of five reinforced members around the frontal
 structure of a car, which can significantly affect the crash safety.
```

---

Car Cab Design (domain knowledge)

```
- A car is subjected to a side-impact crash test. The outcome variables y measure the
following:
    - The effect of the side-impact on a dummy is measured in terms of head injury,
load in abdomen, pubic symphysis force, viscous criterion (V * C), and rib deflections
 at the upper, middle and lower rib locations.
    - The effects on the car are considered in terms of the vehicle's weight, the
velocity of the B-Pillar at the middle point and the velocity of the front door at the
 B-Pillar.
- The parameters x describe some design aspects of the car. An increase in dimension
of the car parameters may improve safety, but with a burden of an increased weight of
the car. These parameters and their ranges are:
    x1: Thickness of B-Pillar inner [0.5, 1.5]
    x2: Thickness of B-Pillar reinforcement [0.45, 1.35]
    x3: Thickness of floor side inner [0.5, 1.5]
    x4: Thickness of cross members [0.5, 1.5]
    x5: Thickness of door beam [0.875, 2.625]
    x6: Thickness of door beltline reinforcement [0.4, 1.2]
    x7: Thickness of roof rail [0.4, 1.2]
- NOTE: The presented values of outcomes y represent the reduction in mass, forces,
velocities etc. So the goal is to increase y_1, ..., y_11, corresponding to lowering
the vehicle's weight and minimizing the impact on the dummy and the car.
```

DM's answer generation for `DM.get_answers`

```
Suppose you are a decision maker evaluating the results of a multi-objective
optimization problem.

You are given a set of multi-dimensional outcomes y = {y_names}

{utility_func_desc}

You have observed the following outcomes with their corresponding utility values and
contributions to the overall utility.

## Outcomes:

{outcomes_markdown}

The utility values are on a scale [0, 1], where (1 - very satisfied, 0.5 - neutral, 0 -
 very dissatisfied).

Based on the above information, provide answers to the following questions:

## Questions:

{questions_str}

Return your final answer as a json file with the following format:
```json
{{
    "q1" : <answer to q1>,
    ...
    "q{n_questions}" : <answer to q{n_questions}>
}}
```
Before providing your final answers, analyze the shape of the utility function in
light of the questions posed.
In your final answers, you cannot reveal the explicit formula of the utility function.
The form and values of the utility functions is a "latent" feature of the human expert,
 thus you should not refer to it explicitly or even mention its existence.
{utility_constraints}
State your answers in the first person (you are the decision maker). Avoid vacuous
statements.
```

*Prompt 9.* The prompt used for answer generation in the `DM.get_answers` subroutine.

# E. Extended Related Work

LILO sits at the intersection of preferential BO, preference elicitation, and recent work on combining LLMs with black-box optimization. **Preferential BO methods** rely on structured feedback formats and do not support rich natural-language descriptions of high-level goals and trade-offs. Existing **LLM-as-optimizer** and **LLM-enhanced BO** methods use LLMs for candidate generation, warm-starting, surrogate modeling, or feature extraction, but generally do not use language as the primary source of preference feedback and often lack principled uncertainty modeling. **Language-based preference elicitation** methods support natural-language interaction but are typically restricted to finite candidate sets rather than continuous black-box optimization. The key novelty of LILO is using natural language as the feedback interface while preserving the uncertainty-aware machinery of BO and supporting textual priors for warm-starting.

*Table 1. Comparison of prior works grouped into methodological families.*

| Family | Method | Feedback | Search Space | LLM Role | Uncertainty modeling | Limitation Relative to LILO |
|---|---|---|---|---|---|---|
| Preferential BO | González et al. (2017) | Pairwise preferences | Continuous | None | GP | Assumes structured feedback such as pairwise comparisons or multi-duels, rather than free-form language feedback. |
| | Lin et al. (2022) | Pairwise preferences | Continuous | None | GP | |
| | Sui et al. (2017) | Multi-duels / best-of-$k$ | Finite arms | None | GP | |
| | Astudillo et al. (2023) | $q$-wise preferences | Continuous | None | GP | |
| LLM-as-Optimizer LLM-Enhanced BO | Yang et al. (2023) | Scalar objective values | Anything text-representable | LLM as optimizer | No | Uses the LLM for search, candidate generation, or feature extraction. Does not use language as the primary source of feedback. Often no principled uncertainty modeling. |
| | Liu et al. (2024) | Scalar objective values | Continuous | Warm-starting, candidate generation, surrogate modeling | LLM | |
| | Ramos et al. (2023) | Scalar objective values | Continuous | Surrogate modeling | LLM | |
| | Cai et al. (2025) | Scalar objective values & Language feedback | Prompt & image editing parameters | Prompt refinement | GP (partial) | |
| | Agarwal et al. (2025) | Scalar objective values | Anything text-representable | Candidate generation & embedding model | GP | |
| | Kristiadi et al. (2024) | Scalar objective values | Molecular design | Feature extractor & embedding model | GP | |
| Language-Based Preference Elicitation | Austin et al. (2024) | Answers to binary NL questions | Finite candidate set | Query verbalization | Parametric model | Focuses on preference elicitation over discrete candidate sets, not continuous black-box optimization with expensive experiments. |
| | Handa et al. (2024) | Answers to binary NL questions | Finite candidate set | Feature extraction & query verbalization | Parametric model | |
| | **LILO (ours)** | Free-form NL | Continuous | Feedback interpretation | GP | – |

# F. Additional Experimental Results

## F.1. Preference Feedback Batch Size Ablation

It is not straightforward to compare the DM's workload necessary to answer natural language questions vs. providing e.g. pairwise comparisons. The DM's workload would depend heavily on the specific questions being asked, and in the case of pairwise comparisons, it would depend on the complexity of the utility function and the kind of outcomes being presented. Experiments presented in this subsection are aimed at understanding how many pairwise comparisons or queries to the ground-truth utility are roughly equivalent to a single message of the DM in natural language. In Figures 9, 10, and 11, we compare the performance of LILO with $B^{\mathrm{pf}} = 1$ with the quantitative baselines (true utility BO and preference BO) with $B^{\mathrm{pf}} \in \{1, 2, 4, 8, 16\}$. The results demonstrate that even one natural language statement can outperform as many as 8-16 pairwise comparisons or point-wise evaluations of the ground-truth utility. As expected, the competitive performance of LILO is most pronounced at the very initial stages of experimentation, which is crucial in feedback-limited regimes.

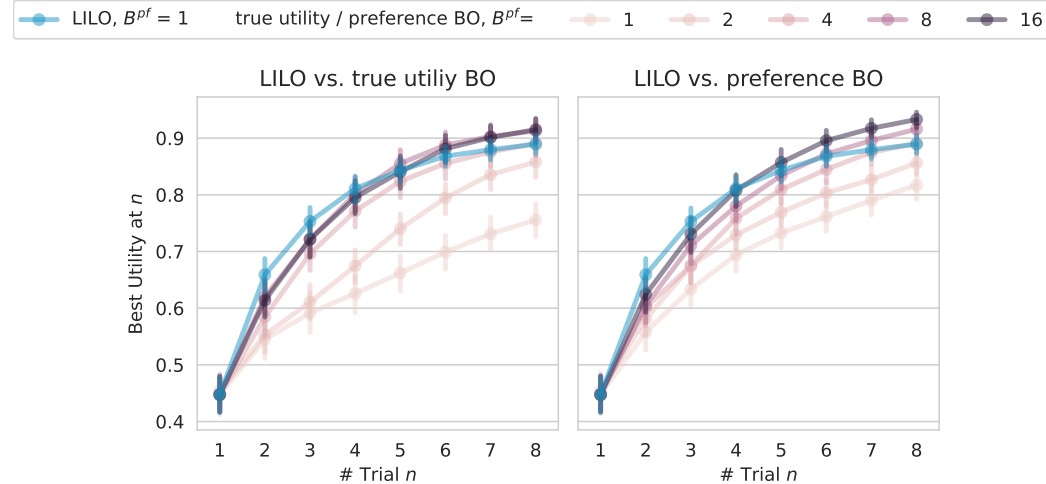

*Figure 9. LILO vs. preference BO and true utility BO with varying feedback batch size.* Results averaged across all environments, with min-max normalization applied within each environment prior to aggregation.

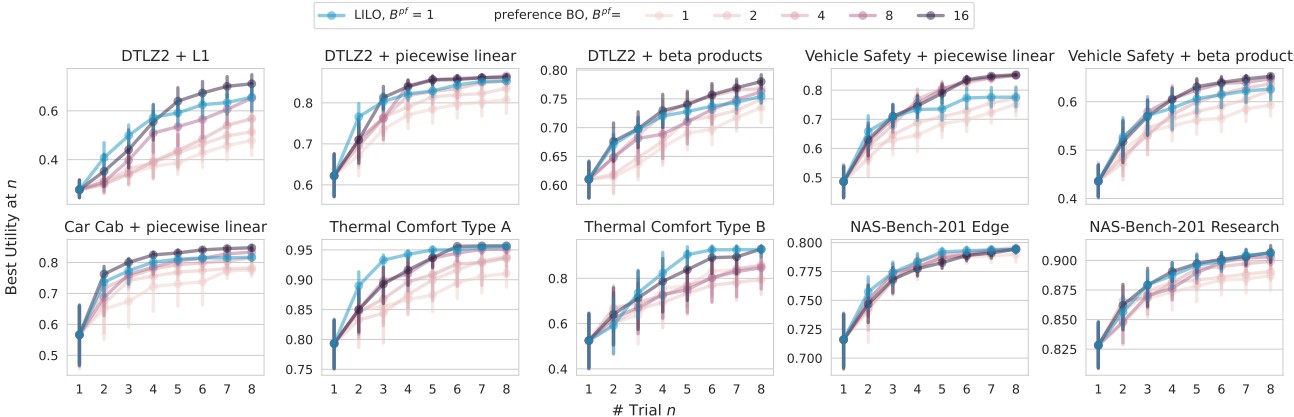

*Figure 10. LILO vs. preference BO with varying feedback batch size.* Results by environment.

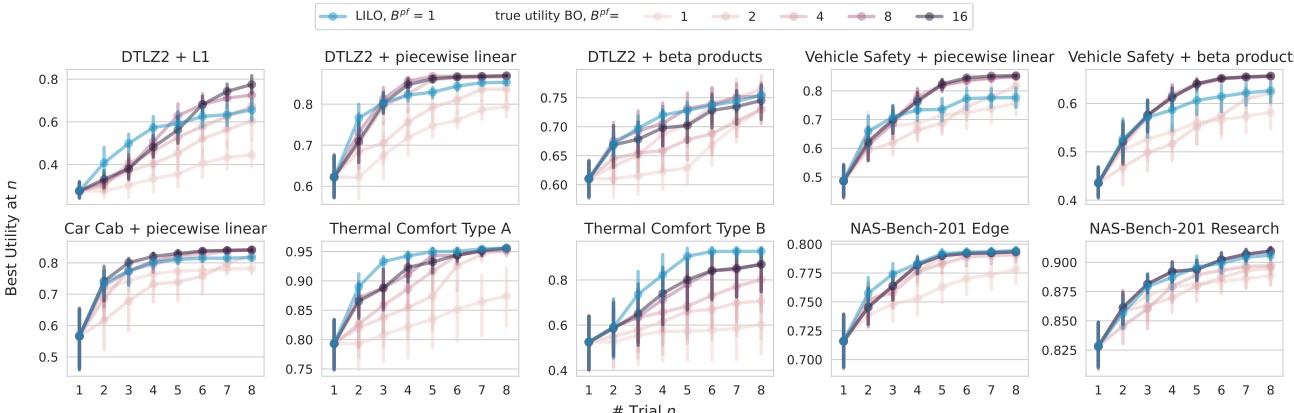

*Figure 11. LILO vs. true utility BO with varying feedback batch size.* Results by environment.

## F.2. Incorporating Prior Knowledge

In addition to the summarized results on prior knowledge incorporation from Section 5.3.2, in Figure 12 we present an environment-by-environment view of these results.

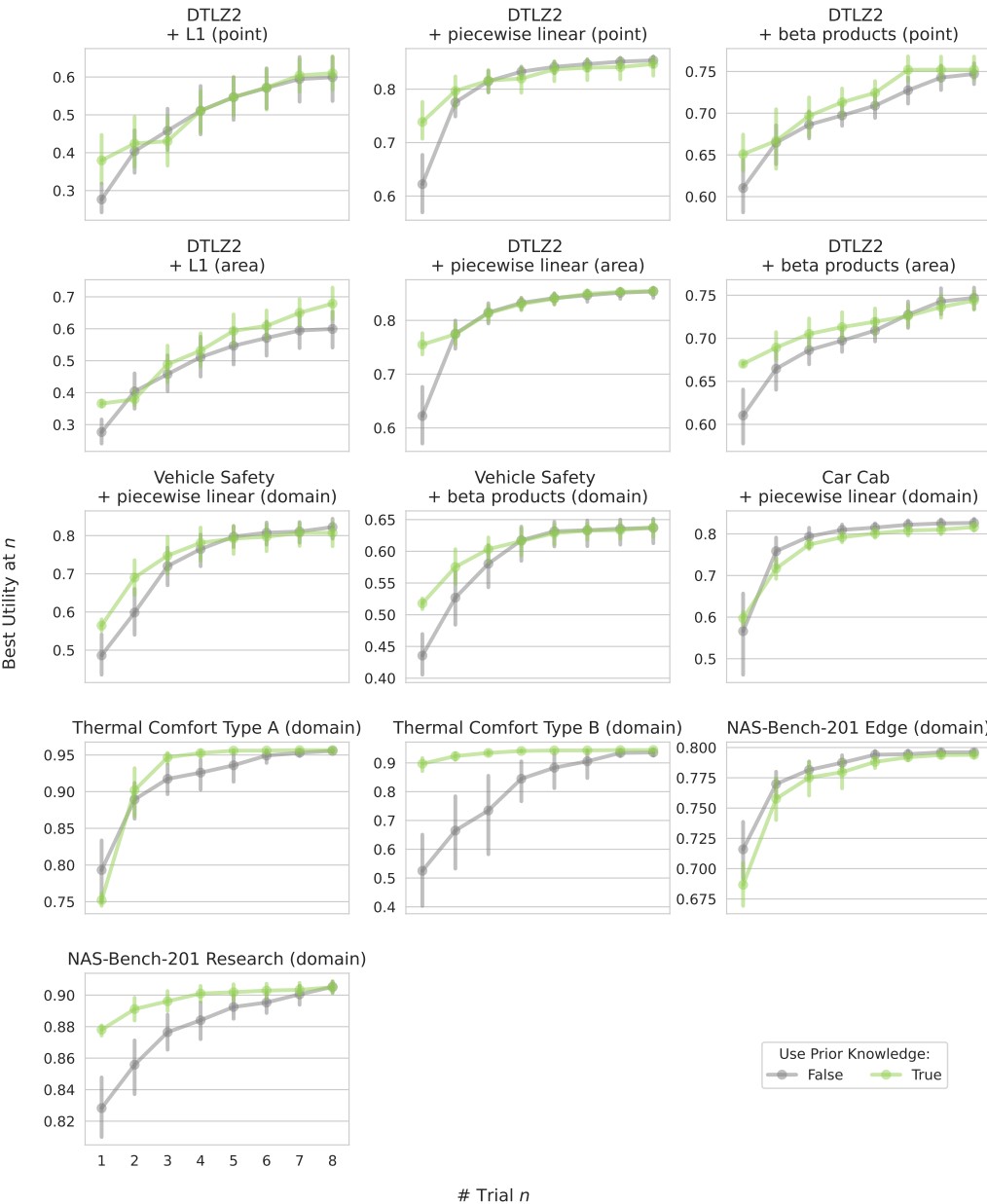

*Figure 12.* Performance of `LILO` with and without prior knowledge. Results across environments and knowledge types.

## F.3. Baselines – feedback acquisition ablation

This section aims to justify the choice of the qEUBO feedback acquisition function by the quantitative baselines.

For true utility BO, we consider the following batch acquisition functions: EUBO, Max Value Entropy Search (Moss et al., 2021), Predictive Entropy Search (Seo et al., 2000) and a random acquisition function. For preference-based BO, we compare acquisition functions designed for preference learning: two versions of the EUBO (batched and greedy) acquisition functions and the pairwise BALD (Houlsby et al., 2011) acquisition function. Batched EUBO is implemented by selecting $q$

points from $D_n^{\text{exp}}$ and creating all possible pairs among them. The number $q$ is set to the smallest integer so that $\frac{q(q-1)}{2} > B^{\text{pf}}$ and a subset of $B^{\text{pf}}$ is used. The greedy EUBO is implemented by first computing the value of EUBO for all pairs (at $q = 2$ the EUBO admits an analytic expression) and selecting the top $B^{\text{pf}}$ pairs.

**Results.** We compute the results across all environments, 32 replications and extend the optimization horizon to $T = 16$ iterations for the results to stabilize. As in the main results, we let $B^{\text{exp}} = d$ and $B^{\text{pf}} = 2$. Table 2 shows the average utility of the best candidate at iteration $n$. While the differences in performance across different feedback acquisition methods are not substantial, we observe a slight advantage of qEUBO against the alternatives. This observation is in line with the prior work on preference-based BO (Lin et al., 2022). These results justify our choice of baseline implementation.

*Table 2. The choice of feedback acquisition function.* Values represent the average of the best ground-truth utility at iteration $n$. Averaged across three simulation environments and 30 replications per environment; min-max standardized within an environment before aggregation. Error bars represent 1 standard deviation of the mean.

*(a)* True Utility BO

| trial $n$ | qEUBO | qMaxES | qPredES | random |
|---|---|---|---|---|
| 1 | 0.449 (0.016) | 0.449 (0.016) | 0.449 (0.016) | 0.449 (0.016) |
| 2 | 0.532 (0.017) | 0.529 (0.017) | 0.532 (0.017) | 0.53 (0.017) |
| 3 | 0.599 (0.017) | 0.604 (0.016) | 0.609 (0.017) | 0.602 (0.017) |
| 4 | 0.667 (0.016) | 0.673 (0.016) | 0.672 (0.016) | 0.659 (0.017) |
| 5 | 0.724 (0.016) | 0.735 (0.016) | 0.73 (0.016) | 0.718 (0.016) |
| 6 | 0.776 (0.016) | 0.781 (0.016) | 0.782 (0.016) | 0.764 (0.017) |
| 7 | 0.809 (0.016) | 0.819 (0.015) | 0.819 (0.016) | 0.793 (0.016) |
| 8 | 0.838 (0.015) | 0.845 (0.015) | 0.848 (0.015) | 0.814 (0.016) |
| 9 | 0.862 (0.014) | 0.867 (0.014) | 0.866 (0.014) | 0.839 (0.015) |
| 10 | 0.879 (0.014) | 0.88 (0.014) | 0.877 (0.014) | 0.854 (0.015) |
| 11 | 0.89 (0.014) | 0.889 (0.013) | 0.885 (0.013) | 0.863 (0.015) |
| 12 | 0.905 (0.013) | 0.898 (0.013) | 0.89 (0.013) | 0.871 (0.014) |
| 13 | 0.913 (0.012) | 0.903 (0.013) | 0.894 (0.013) | 0.879 (0.014) |
| 14 | 0.917 (0.012) | 0.908 (0.013) | 0.899 (0.013) | 0.885 (0.014) |
| 15 | 0.923 (0.012) | 0.911 (0.013) | 0.903 (0.013) | 0.892 (0.013) |
| 16 | 0.928 (0.011) | 0.916 (0.012) | 0.906 (0.013) | 0.897 (0.013) |

*(b)* Preference BO

| trial $n$ | qEUBO | qEUBO (greedy) | BALD | random |
|---|---|---|---|---|
| 1 | 0.45 (0.016) | 0.45 (0.016) | 0.45 (0.016) | 0.45 (0.016) |
| 2 | 0.567 (0.016) | 0.567 (0.017) | 0.568 (0.017) | 0.566 (0.017) |
| 3 | 0.641 (0.016) | 0.649 (0.016) | 0.636 (0.016) | 0.627 (0.016) |
| 4 | 0.7 (0.015) | 0.696 (0.015) | 0.699 (0.016) | 0.691 (0.016) |
| 5 | 0.746 (0.015) | 0.748 (0.014) | 0.741 (0.015) | 0.739 (0.015) |
| 6 | 0.795 (0.013) | 0.79 (0.012) | 0.774 (0.014) | 0.77 (0.014) |
| 7 | 0.822 (0.012) | 0.814 (0.012) | 0.809 (0.013) | 0.803 (0.013) |
| 8 | 0.844 (0.012) | 0.843 (0.011) | 0.836 (0.012) | 0.828 (0.013) |
| 9 | 0.865 (0.011) | 0.864 (0.011) | 0.854 (0.011) | 0.842 (0.012) |
| 10 | 0.879 (0.01) | 0.879 (0.01) | 0.871 (0.011) | 0.861 (0.011) |
| 11 | 0.893 (0.01) | 0.893 (0.01) | 0.885 (0.01) | 0.875 (0.011) |
| 12 | 0.907 (0.008) | 0.902 (0.009) | 0.9 (0.008) | 0.885 (0.01) |
| 13 | 0.917 (0.008) | 0.912 (0.009) | 0.912 (0.008) | 0.9 (0.009) |
| 14 | 0.927 (0.007) | 0.918 (0.009) | 0.923 (0.007) | 0.909 (0.009) |
| 15 | 0.934 (0.006) | 0.922 (0.008) | 0.933 (0.007) | 0.913 (0.009) |
| 16 | 0.939 (0.006) | 0.928 (0.008) | 0.939 (0.006) | 0.919 (0.009) |

## F.4. LLM's Accuracy

In this section, we look at the accuracy of LILO in generating pairwise preference labels. Figure 13 shows the average accuracy at each trial, computed across 16 seeds and the 3 DTLZ2 environments. We find that already at the first iteration, LILO yields high-fidelity predictions, with an average accuracy of 85%. As more information about the optimization objective is gathered, its accuracy reaches values above 90%. The relatively high accuracy contrasted with low labeling costs (in comparison to the cost of human labor) justifies the validity of our approach – human feedback in natural language can be effectively translated into a numerical signal suitable for optimization via LLM pairwise labeling followed by GP model fitting.

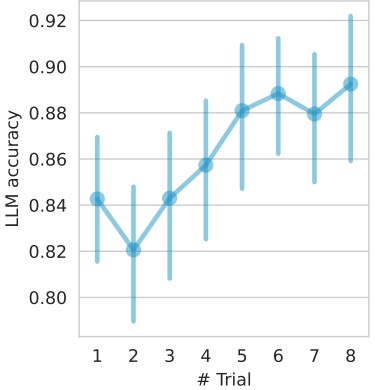

*Figure 13.* The accuracy of LILO in generating pairwise preference choices in step 3 of the BO loop.

## F.5. Longer Trials

We note that this paper primarily concerns settings where configurations are very costly to evaluate, and therefore in practice the number of trials is very limited. Our overarching goal is to minimize human effort during optimization, which is why the main experiments focus on the impact of queries after only a few batched rounds. Nevertheless, in this section, we present additional results evaluating the performance of LILO over longer horizons, with up to 16 batched experimental trials. Figure 14 compares LILO to both preferential and true utility BO (excluding the two LLM baselines, as they do not show meaningful progression across iterations). We observe that LILO maintains competitive performance even in this extended setting. As noted in the main results, the advantage of LILO tends to diminish as the number of experimental trials increases, with quantitative baselines eventually catching up and, in some cases, surpassing LILO in the long run. This behavior is expected – the in-context learning capabilities of LLMs are inherently limited and do not offer the convergence guarantees provided by conventional baselines operating on quantitative feedback.

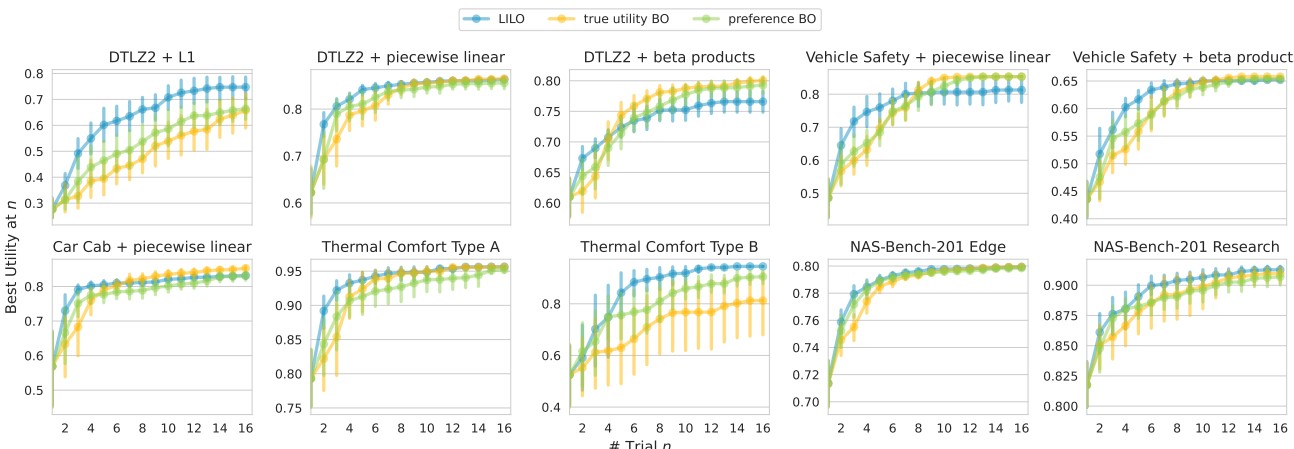

*Figure 14. LILO vs. true utility and preference BO on longer horizons.* Results aggregated across 16 replications per environment.

### F.6. Acquisition Function Ablation

In our main experiments, we demonstrate the performance of `LILO` against baselines with all methods using the (Noisy) Expected Improvement as the acquisition function for candidate generation. However, this choice can be replaced with other alternatives. In this section, we demonstrate that `LILO` maintains competitive performance irrespective of this choice.

In Figures 15 and 16 we demonstrate the performance of `LILO` and the quantitative baselines with three different choices of acquisition functions for candidate generation: (Noisy) Expected Improvement (as in the main text), Upper Confidence Bound (UCB) with the parameter $\beta$ controlling the trade-off between exploration and exploitation set to 0.5, and Thompson Sampling. All other elements of `LILO` and the baselines stay fixed. Figure 15 shows results summarized across all environments considered and Figure 16 shows a detailed view of the results. We observe that the Expected Improvement and the UCB acquisition functions, overall, perform better than Thompson Sampling on our selected set of test problems. The advantage of `LILO` against the baselines is most competitive in these two settings. With Thompson Sampling, `LILO` performs similarly to the true utility BO baseline, slightly outperforming it at the very first iterations.

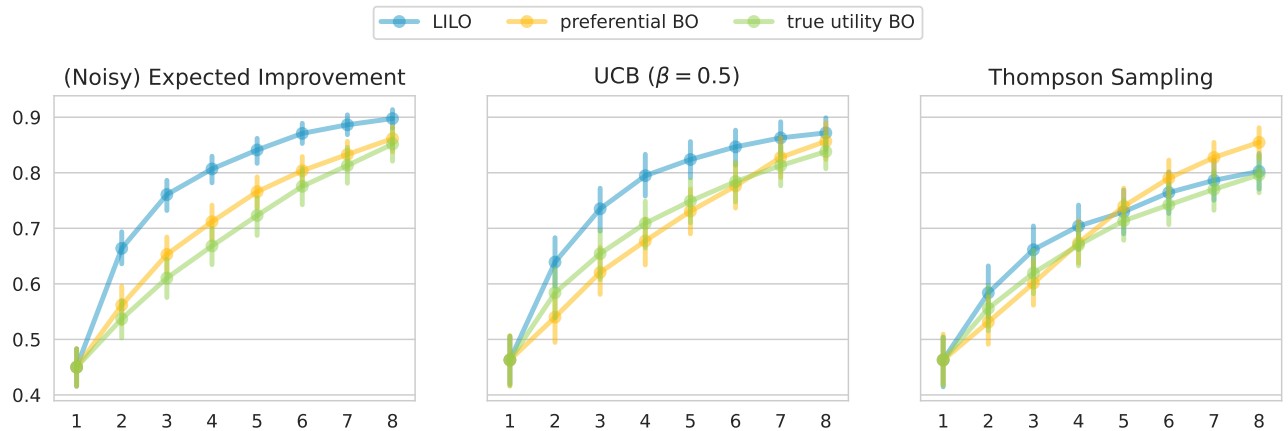

*Figure 15. LILO vs. quantitative baselines using different acquisition functions for candidate generation.* Results averaged across all 10 environments with 16 replications per environment.

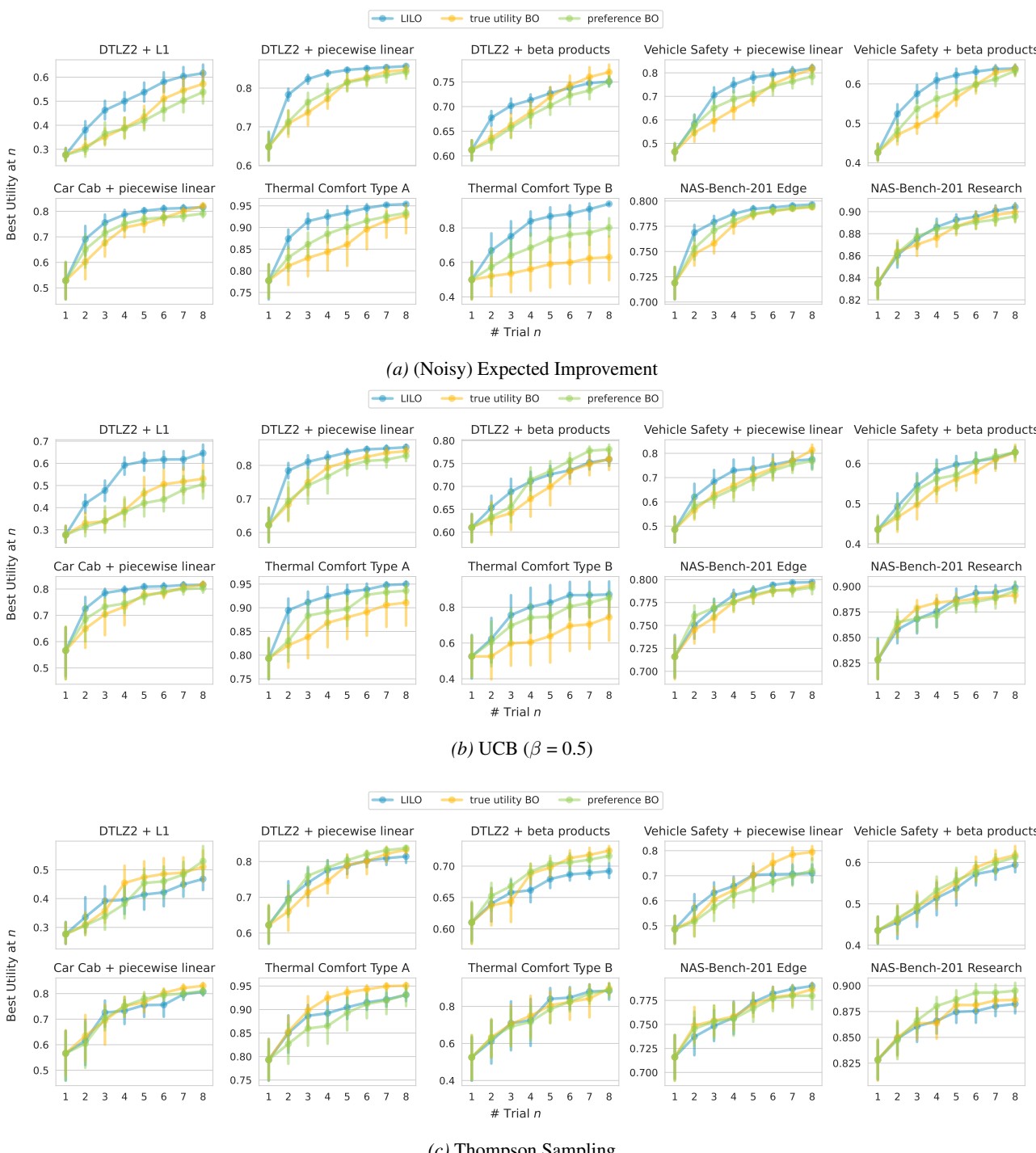

*Figure 16. LILO vs. quantitative baselines using different acquisition functions for candidate generation.* Detailed results across all 10 environments with 16 replications per environment.

## F.7. LLM Ablation Study

We compare the performance of `LILO` depending on the choice of the LLM model. We compare the following models: Llama-3.3-70b-instruct, Llama-4-scout-17b-16e-instruct, and Qwen3-14B. The LLM used to simulate human feedback remains set to Llama-3.3-70b-instruct across the comparisons.

We run the ablation study for the following 3 environments: the DTLZ2 outcome function, combined with the L1, beta products and piecewise linear utility functions. As in the main benchmark, we set $B^{\mathrm{pf}} = 2$ and $B^{\mathrm{exp}} = d = 8$.

**Results.** Table 3 presents the results. We observe that `LILO` performs similarly across all three LLMs, demonstrating that the success of our method is agnostic to the choice of a specific language model. We were unable to test the performance of `LILO` with smaller language models (e.g. Llama-3.1-8B or Qwen3-8B) due to difficulties in ensuring that the LLM's outputs follow the required json structure, leading to parsing errors.

*Table 3. LLM ablation study.* Max value of the ground-truth utility achieved after $n$ iterations. Error bars are 1 standard deviation of the mean across 30 simulation replications.

*(a)* DTLZ2 + L1

| method
# trial | LILO
(Llama-3.3-70b) | LILO
(Llama-4-scout) | LILO
(Qwen3-14B) | preference BO | true utility BO |
|---|---|---|---|---|---|
| 1 | 0.28 ± 0.01 | 0.28 ± 0.01 | 0.28 ± 0.01 | 0.28 ± 0.01 | 0.28 ± 0.01 |
| 2 | 0.42 ± 0.02 | 0.43 ± 0.02 | 0.42 ± 0.02 | 0.31 ± 0.02 | 0.33 ± 0.02 |
| 3 | 0.54 ± 0.03 | 0.53 ± 0.02 | 0.54 ± 0.03 | 0.35 ± 0.02 | 0.37 ± 0.02 |
| 4 | 0.56 ± 0.03 | 0.59 ± 0.02 | 0.59 ± 0.03 | 0.37 ± 0.02 | 0.4 ± 0.02 |
| 5 | 0.62 ± 0.02 | 0.64 ± 0.02 | 0.63 ± 0.03 | 0.41 ± 0.02 | 0.43 ± 0.02 |
| 6 | 0.64 ± 0.02 | 0.67 ± 0.02 | 0.66 ± 0.03 | 0.44 ± 0.02 | 0.46 ± 0.02 |
| 7 | 0.67 ± 0.02 | 0.7 ± 0.02 | 0.69 ± 0.02 | 0.46 ± 0.02 | 0.5 ± 0.02 |
| 8 | 0.69 ± 0.02 | 0.71 ± 0.02 | 0.71 ± 0.02 | 0.5 ± 0.02 | 0.54 ± 0.03 |

*(b)* DTLZ2 + beta products

| method
# trial | LILO
(Llama-3.3-70b) | LILO
(Llama-4-scout) | LILO
(Qwen3-14B) | preference BO | true utility BO |
|---|---|---|---|---|---|
| 1 | 0.61 ± 0.01 | 0.61 ± 0.01 | 0.61 ± 0.01 | 0.61 ± 0.01 | 0.61 ± 0.01 |
| 2 | 0.66 ± 0.01 | 0.67 ± 0.01 | 0.66 ± 0.01 | 0.63 ± 0.01 | 0.63 ± 0.01 |
| 3 | 0.69 ± 0.01 | 0.69 ± 0.01 | 0.69 ± 0.01 | 0.66 ± 0.01 | 0.66 ± 0.01 |
| 4 | 0.72 ± 0.01 | 0.71 ± 0.01 | 0.7 ± 0.01 | 0.68 ± 0.01 | 0.69 ± 0.01 |
| 5 | 0.73 ± 0.0 | 0.73 ± 0.01 | 0.71 ± 0.01 | 0.69 ± 0.01 | 0.72 ± 0.01 |
| 6 | 0.74 ± 0.01 | 0.74 ± 0.01 | 0.73 ± 0.01 | 0.71 ± 0.01 | 0.74 ± 0.01 |
| 7 | 0.75 ± 0.01 | 0.75 ± 0.01 | 0.73 ± 0.01 | 0.72 ± 0.01 | 0.76 ± 0.01 |
| 8 | 0.76 ± 0.01 | 0.75 ± 0.0 | 0.73 ± 0.01 | 0.74 ± 0.01 | 0.76 ± 0.01 |

*(c)* DTLZ2 + piecewise linear

| method
# trial | LILO
(Llama-3.3-70b) | LILO
(Llama-4-scout) | LILO
(Qwen3-14B) | preference BO | true utility BO |
|---|---|---|---|---|---|
| 1 | 0.65 ± 0.02 | 0.65 ± 0.02 | 0.65 ± 0.02 | 0.65 ± 0.02 | 0.65 ± 0.02 |
| 2 | 0.77 ± 0.01 | 0.76 ± 0.01 | 0.76 ± 0.01 | 0.68 ± 0.02 | 0.71 ± 0.02 |
| 3 | 0.8 ± 0.01 | 0.82 ± 0.01 | 0.82 ± 0.01 | 0.74 ± 0.01 | 0.76 ± 0.02 |
| 4 | 0.83 ± 0.01 | 0.83 ± 0.01 | 0.84 ± 0.01 | 0.77 ± 0.02 | 0.78 ± 0.01 |
| 5 | 0.85 ± 0.0 | 0.84 ± 0.01 | 0.85 ± 0.0 | 0.8 ± 0.01 | 0.81 ± 0.01 |
| 6 | 0.85 ± 0.0 | 0.85 ± 0.01 | 0.85 ± 0.0 | 0.81 ± 0.01 | 0.83 ± 0.01 |
| 7 | 0.86 ± 0.0 | 0.85 ± 0.0 | 0.86 ± 0.0 | 0.81 ± 0.01 | 0.85 ± 0.01 |
| 8 | 0.86 ± 0.0 | 0.86 ± 0.0 | 0.86 ± 0.0 | 0.83 ± 0.01 | 0.86 ± 0.0 |

### F.8. High-dimensional and Unstructured Outcomes

In addition to the synthetic and structured low-dimensional environments considered in the main paper, we evaluate `LILO` in a setting where the realized outcomes are *textual summaries*. This example is meant to illustrate a key property of the method: `LILO` does not require an explicit surrogate model over the outcome space $\mathcal{Y}$ itself. Instead, it only requires that an LLM can interpret realized outcomes and infer pairwise preferences over them given natural-language feedback. This makes the framework naturally compatible with high-dimensional and unstructured outcomes such as text or images.

#### F.8.1. ENVIRONMENT

We consider abstractive summarization on the CNN/DailyMail dataset (Hermann et al., 2015; See et al., 2017). For a given article $a$, a summarization LLM produces a summary

$$s = f(a, \mathbf{x}),$$

where the controllable input

$$\mathbf{x} = (x_1, x_2, x_3, x_4)$$

consists of four decoding hyperparameters: temperature, top-$p$, maximum output length, and repetition penalty.

Unlike the other environments, the realized outcome is not a low-dimensional numeric vector but the generated summary text itself. To define an oracle utility for evaluation, we associate each generated summary with a set of interpretable quality indicators computed relative to the article and the gold reference summary provided in the dataset.

#### F.8.2. LATENT UTILITY FUNCTION

For a generated summary $s$, article $a$, and reference summary $r$, we compute the following quality indicators:

$$
\begin{aligned}
q_{\text{R1}}(s, r) &:= \text{ROUGE-1 F1}(s, r), \\
q_{\text{R2}}(s, r) &:= \text{ROUGE-2 F1}(s, r), \\
q_{\text{RL}}(s, r) &:= \text{ROUGE-L F1}(s, r), \\
q_{\text{read}}(s) &:= \text{readability score}, \\
q_{\text{conc}}(s) &:= \text{conciseness score}, \\
q_{\text{rep}}(s) &:= \text{non-repetition score}, \\
q_{\text{comp}}(s) &:= \text{completion score}.
\end{aligned}
$$

Here, ROUGE-1, ROUGE-2, and ROUGE-L are standard summarization metrics based on overlap with a reference summary (Lin, 2004). ROUGE-1 measures unigram overlap, ROUGE-2 bigram overlap, and ROUGE-L overlap based on the longest common subsequence. The readability score is obtained from normalized Flesch Reading Ease (Flesch, 1948), which rewards shorter sentences and simpler vocabulary. The conciseness score softly favors summaries of approximately 90 words, the non-repetition score rewards lexical diversity, and the completion score is binary:

$$q_{\text{comp}}(s) = \begin{cases} 1, & \text{if } s \text{ ends with a complete sentence}, \\ 0, & \text{otherwise}. \end{cases}$$

We define the latent utility as

$$u(s; a, r) = (w_{\text{R1}} q_{\text{R1}} + w_{\text{R2}} q_{\text{R2}} + w_{\text{RL}} q_{\text{RL}} + w_{\text{read}} q_{\text{read}} + w_{\text{conc}} q_{\text{conc}} + w_{\text{rep}} q_{\text{rep}}) \cdot \max(q_{\text{comp}}, 0.1).$$

In all experiments we use the fixed weights

$$(w_{\text{R1}}, w_{\text{R2}}, w_{\text{RL}}, w_{\text{read}}, w_{\text{conc}}, w_{\text{rep}}) = (0.25, 0.25, 0.20, 0.15, 0.10, 0.05).$$

This choice puts the strongest emphasis on content overlap with the reference summary while still rewarding readability, conciseness, and non-repetition. The multiplicative factor $\max(q_{\text{comp}}, 0.1)$ strongly penalizes summaries that terminate mid-sentence without collapsing their utility exactly to zero.

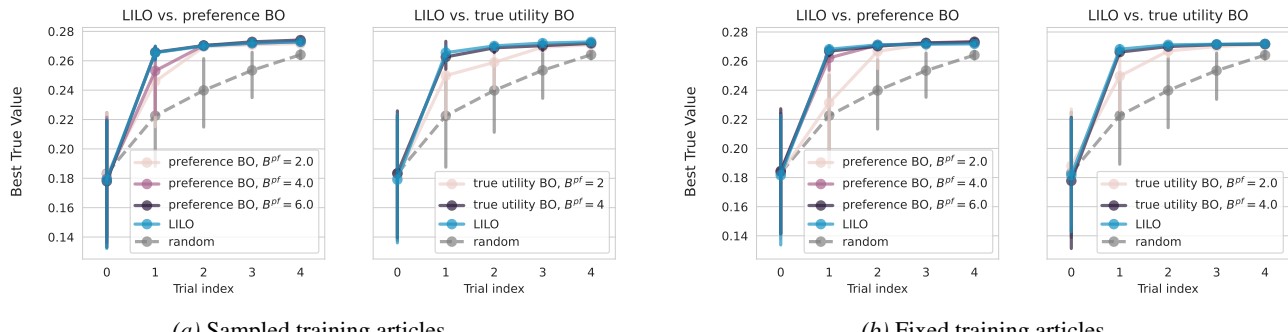

*(a)* Sampled training articles.                    *(b)* Fixed training articles.

*Figure 17. Optimization results on the text summarization benchmark.* In the sampled setting, each BO iteration uses a fresh batch of $N^{\mathrm{art}} = 3$ training articles. In the fixed setting, all BO iterations reuse the same set of $N^{\mathrm{art}} = 10$ training articles and the natural-language feedback is accumulated additively across rounds. In both cases, each trial evaluates $N^{\mathrm{cfg}} = 4$ candidate decoding configurations and optimization is run for $T = 5$ batched trials. Final evaluation is performed on a held-out test set of 20 unseen articles. Preference BO is evaluated with feedback budgets $B^{\mathrm{pf}} \in \{2, 4, 6\}$, while true-utility BO uses $B^{\mathrm{pf}} \in \{2, 4\}$. Across both settings, LILO matches the strongest baselines when those baselines are given the maximum amount of oracle feedback, and for smaller feedback budgets it typically improves more rapidly in the first one or two optimization rounds.

### F.8.3. OPTIMIZATION PROTOCOL

At each BO iteration, a batch of candidate decoding configurations is evaluated on a batch of articles. For every configuration–article pair, the summarization LLM generates a summary, the associated quality indicators are computed, and a DM LLM produces natural-language feedback describing which summaries it prefers and why. The LILO agent then converts this feedback into pairwise labels between candidate summaries and uses the resulting comparisons to update the GP proxy model over decoding parameters. We consider two article-selection strategies.

**Sampled articles.** At each BO trial, a fresh minibatch of training articles is sampled from $\mathcal{D}_{\mathrm{train}}$. Natural-language feedback is generated independently for that batch, and pairwise labels are inferred only from the current summaries and critiques. This protocol is summarized in Algorithm 6.

**Fixed articles.** A fixed set of training articles is reused across all BO trials. In this setting, the natural-language feedback is accumulated *additively*: rather than regenerating the critique from scratch, the DM is shown the previously generated feedback together with the newly generated summaries and produces an incremental update. Pairwise preferences are then re-generated from the accumulated feedback. This protocol is summarized in Algorithm 7.

The fixed-article setting provides a more stable decision context across trials, while the sampled-article setting is closer to a stochastic minibatch approximation. In both cases, final evaluation is performed on a held-out set of unseen test articles.

### F.8.4. NATURAL-LANGUAGE FEEDBACK AND PAIRWISE JUDGING

To generate feedback, we use a dedicated LLM agent that is shown: (i) the original article, (ii) the candidate summaries for that article, and (iii) the corresponding utility indicators. The agent is instructed to use the summaries themselves as the main signal and the numeric indicators only as supporting evidence. It is also explicitly told to align its preferences with the latent utility and to strongly penalize incomplete summaries. The feedback-generation prompts used in the standard and incremental settings are shown in Prompts 10 and 11.

A second LLM agent acts as the pairwise judge. It is given the article, the reference summary, the natural-language feedback, and the candidate summaries, and is asked to return pairwise preferences between the summaries in JSONL format. The exact prompt is shown in Prompt 12.

### F.8.5. RESULTS

The empirical results are shown in Figure 17. In the sampled setting, each BO iteration uses a fresh batch of $N^{\mathrm{art}} = 3$ training articles. In the fixed setting, all BO trials reuse the same set of $N^{\mathrm{art}} = 10$ training articles and the feedback is accumulated additively across rounds. In both cases, each trial evaluates $N^{\mathrm{cfg}} = 4$ candidate decoding configurations,

matching the dimensionality of the search space, and optimization is run for $T = 5$ batched trials, after which all methods have effectively saturated. Final performance is evaluated on a held-out test set of 20 unseen articles.

With $N^{\text{cfg}} = 4$, each BO trial contains at most six unique pairwise comparisons and four direct utility evaluations. Accordingly, preference BO is evaluated with feedback budgets $B^{\text{pf}} \in \{2, 4, 6\}$, while true-utility BO is evaluated with $B^{\text{pf}} \in \{2, 4\}$. We also report a random acquisition baseline.

Across both the sampled and fixed settings, LILO matches the strongest baselines when those baselines are given the maximum amount of oracle feedback. Since preference BO with $B^{\text{pf}} = 6$ and true-utility BO with $B^{\text{pf}} = 4$ effectively represent upper bounds on the amount of information available per trial, this indicates that the natural-language feedback and pairwise-labeling pipeline is able to recover highly informative preference information. For smaller feedback budgets, LILO typically improves more rapidly during the first one to two optimization rounds. All BO-based methods substantially outperform random search.

Overall, these results show that LILO can optimize over high-dimensional textual outcomes without explicitly modeling the outcome space itself. Instead, it relies only on natural-language critiques and pairwise judgments over generated summaries, while achieving optimization performance comparable to methods with direct access to the oracle latent utility function.

---

**Algorithm 6:** LILO with **sampled** articles

**Input:** Number of BO trials $T$, number of candidate configurations per trial $N^{\text{cfg}}$, number of articles per trial $N^{\text{art}}$, training article pool $\mathcal{D}_{\text{train}}$, test article set $\mathcal{D}_{\text{test}}$

**Output:** Proxy model $M_T$ and final incumbent decoding configuration

$D_0^{\text{exp}} \leftarrow \emptyset$
$D_0^{\text{pf}} \leftarrow \emptyset$
**for** $n = 1$ **to** $T$ **do**
    **if** $n = 1$ **then**
        $\{x_i\}_{i=1}^{N^{\text{cfg}}} \sim \text{Uniform}(\mathcal{X})$
    **else**
        $\{x_i\}_{i=1}^{N^{\text{cfg}}} \sim \text{acqf}(\mathcal{X}; M_{n-1})$
    **end**
    $\{a_j\}_{j=1}^{N^{\text{art}}} \sim \mathcal{D}_{\text{train}}$
    **foreach** $x_i \in \{x_i\}_{i=1}^{N^{\text{cfg}}}$ and $a_j \in \{a_j\}_{j=1}^{N^{\text{art}}}$ **do**
        Generate summary $s_{ij} = f(a_j, x_i)$
        Compute summary indicators $q_{ij}$ and latent utility $u_{ij}$
    **end**
    $D_n^{\text{exp}} \leftarrow D_{n-1}^{\text{exp}} \cup \{(x_i, a_j, s_{ij}, q_{ij})\}_{i=1, j=1}^{N^{\text{cfg}}, N^{\text{art}}}$
    **foreach** $a_j \in \{a_j\}_{j=1}^{N^{\text{art}}}$ **do**
        $F_j^{(n)} \leftarrow \text{DM.get\_feedback}\Big(a_j, \{s_{ij}, q_{ij}\}_{i=1}^{N^{\text{cfg}}}\Big)$
        $P_j^{(n)} \leftarrow \text{LILO.get\_pair\_labels}\Big(a_j, r_j, \{s_{ij}\}_{i=1}^{N^{\text{cfg}}}, F_j^{(n)}\Big)$
    **end**
    $D_n^{\text{pf}} \leftarrow D_{n-1}^{\text{pf}} \cup \{(a_j, P_j^{(n)})\}_{j=1}^{N^{\text{art}}}$
    $M_n \leftarrow \text{fit proxy}\big(D_n^{\text{exp}}, D_n^{\text{pf}}\big)$
**end**

---

*Figure 18.* LILO with **sampled** articles. At each BO trial, a fresh batch of training articles is sampled, candidate summaries are generated, natural-language feedback is collected, pairwise labels are inferred, and the proxy model is updated.

---

**Algorithm 7:** `LILO` with **fixed** articles

---

**Input:** Number of BO trials $T$, number of candidate configurations per trial $N^{\text{cfg}}$, fixed training article set $\mathcal{A}_{\text{train}} = \{a_j\}_{j=1}^{N^{\text{art}}}$,
      test article set $\mathcal{D}_{\text{test}}$
**Output:** Proxy model $M_T$ and final incumbent decoding configuration
$D_0^{\text{exp}} \leftarrow \emptyset$
$D_0^{\text{pf}} \leftarrow \emptyset$
$\mathcal{F}_j \leftarrow \emptyset$ for all $a_j \in \mathcal{A}_{\text{train}}$
**for** $n = 1$ **to** $T$ **do**
    **if** $n = 1$ **then**
        $\{x_i\}_{i=1}^{N^{\text{cfg}}} \sim \text{Uniform}(\mathcal{X})$
    **else**
        $\{x_i\}_{i=1}^{N^{\text{cfg}}} \sim \text{acqf}(\mathcal{X}; M_{n-1})$
    **end**

    **foreach** $x_i \in \{x_i\}_{i=1}^{N^{\text{cfg}}}$ *and* $a_j \in \mathcal{A}_{\text{train}}$ **do**
        Generate summary $s_{ij}^{(n)} = f(a_j, x_i)$
        Compute summary indicators $q_{ij}^{(n)}$ and latent utility $u_{ij}^{(n)}$
    **end**
    $D_n^{\text{exp}} \leftarrow D_{n-1}^{\text{exp}} \cup \{(x_i, a_j, s_{ij}^{(n)}, q_{ij}^{(n)})\}_{i=1, a_j \in \mathcal{A}_{\text{train}}}$
    **foreach** $a_j \in \mathcal{A}_{\text{train}}$ **do**
        **if** $n = 1$ **then**
            $F_j^{(1)} \leftarrow \text{DM.get\_feedback}\Big(a_j, \{s_{ij}^{(1)}, q_{ij}^{(1)}\}_{i=1}^{N^{\text{cfg}}}\Big)$
        **else**
            $F_j^{(n)} \leftarrow \text{DM.get\_incremental\_feedback}\Big(a_j, \mathcal{F}_j, \{s_{ij}^{(n)}, q_{ij}^{(n)}\}_{i=1}^{N^{\text{cfg}}}\Big)$
        **end**
        $\mathcal{F}_j \leftarrow \mathcal{F}_j \oplus F_j^{(n)}$
        $P_j^{(n)} \leftarrow \text{LILO.get\_pair\_labels}\Big(a_j, r_j, \{s_{ij}^{(\ell)}\}_{i, \ell \leq n}, \mathcal{F}_j\Big)$
    **end**
    $D_n^{\text{pf}} \leftarrow \{(a_j, P_j^{(n)})\}_{a_j \in \mathcal{A}_{\text{train}}}$
    $M_n \leftarrow \text{fit proxy}\big(D_n^{\text{exp}}, D_n^{\text{pf}}\big)$
**end**

---

*Figure 19.* `LILO` with **fixed** training articles. The same set of $N^{\text{art}}$ training articles is reused across all BO trials. At each trial, $N^{\text{cfg}}$ decoding configurations are evaluated on this fixed article set, natural-language feedback is accumulated additively across rounds, pairwise preferences are re-generated from the updated feedback, and the proxy model $M_n$ is refit.

---

Feedback generation in `DM.get_feedback`

---

```
You are simulating a human decision-maker giving feedback on candidate summaries of a
news article.

Your preferences:
- Capturing the key facts from the article is my top priority, coverage and
faithfulness matter most
- I also value readability and conciseness. A good summary should be tight
- roughly a short paragraph. Extra details beyond the core facts hurt more than they
help.
- I dislike repetitive, verbose, or incomplete summaries.

You are shown:
1. The original article
2. Several candidate summaries
3. Some automatically computed quality indicators for each summary

Important:
- Use the summaries themselves as the main signal.
- Use the numeric indicators as supporting evidence.
- Make sure your preferences align with the "overall_utility" score (higher is better,
 range [0, 1]).
- Penalize summaries that finish mid-sentence (the binary "completion" score).
Incomplete summaries receive a high penalty.
- Do not mention any scores explicitly in your feedback.

Instructions:
- First, briefly describe what properties matter most to you for a good summary of
this particular article.
- When evaluating informativeness, focus on whether summaries capture the most
essential facts -- the kind of information that would appear in a short headline-style
 reference summary. Extra details from the article that go beyond the core facts are
less important to you.
- Note any trade-offs you see between factual coverage of core points, readability,
and conciseness.
- Give one or two concrete examples to illustrate. If some summaries are close in
quality but differ in specific ways, mention those differences -- even small ones.
- If you have a slight preference between two similar candidates, say so briefly and
explain why.
- Be concise. Do not write more than a short paragraph.

Original article:
\"\"\"
{article}
\"\"\"

Candidate summaries and indicators:
{summaries_block}

Please provide feedback on the summaries. Describe what properties you found most
important for summarizing this particular article, and what trade-offs you noticed
across the batch.

Return plain text only.
```

*Prompt 10.* Natural-language feedback-generation prompt used in `DM.get_feedback` for text summarization.

---

Feedback generation in `DM.get_incremental_feedback`

```
You are simulating a human decision-maker giving iterative feedback on candidate
summaries of a news article.

Your preferences:
- Capturing the key facts from the article is my top priority, coverage and
faithfulness matter most
- I also value readability and conciseness. A good summary should be tight
- roughly a short paragraph. Extra details beyond the core facts hurt more than they
help.
- I dislike repetitive, verbose, or incomplete summaries.

You previously reviewed several candidate summaries and provided this feedback:
\"\"\"
{previous_feedback}
\"\"\"

Your previous best was candidate_{best_prev_idx}:
\"\"\"
{best_prev_summary}
\"\"\"

You are now shown {len(new_summaries)} new candidate summaries for the same article.

Important:
- Use the summaries themselves as the main signal.
- Use the numeric indicators as supporting evidence.
- Make sure your preferences align with the "overall_utility" score (higher is better,
 range [0, 1]).
- Penalize summaries that finish mid-sentence (the binary "completion" score).
Incomplete summaries receive a high penalty.
- Do not mention any scores explicitly in your feedback.

Instructions:
- Mention explicitly at most two new candidate summaries.
- Compare those new candidates to your previous best (candidate_{best_prev_idx}).
- Be specific about what improved or regressed.
- Be concise. Do not write more than a short paragraph.

Original article:
\"\"\"
{article}
\"\"\"

New candidate summaries and indicators:
{new_summaries_block}

Provide a concise update to your assessment. Focus on what changed.

Return plain text only.
```

*Prompt 11.* Incremental natural-language feedback prompt used in `DM.get_incremental_feedback` for the fixed-article setting.

```
LILO pairwise preference labeling in LILO.get_pair_labels
```

```
You are an expert judge inferring a decision maker's preferences over candidate
summaries.

Your task is to compare candidate summaries PAIRWISE and decide which summary the
decision maker would prefer, based on:
1. the original article,
2. the reference summary,
3. the expert feedback describing the decision maker's preferences.

Important instructions:
- Judge from the perspective of the decision maker described in the feedback.
- If the feedback explicitly praises or criticizes a specific candidate, treat that as
 a strong signal -- it directly reflects the decision maker's preferences.
- Use the original article and reference summary to assess coverage, faithfulness,
readability, conciseness, and completeness.
- Prefer summaries that better match the stated preferences in the feedback.
- For each pair, choose exactly one winner. Do not output ties.
- Compare each pair independently.
- Keep your reasoning short, specific, and grounded in the actual summaries and the
expert feedback.

Original article:
\"\"\"
{article}
\"\"\"

Reference summary:
\"\"\"
{ref_summary}
\"\"\"

Expert feedback expressing the decision maker's preferences:
\"\"\"
{feedback}
\"\"\"

Candidate summaries:
{candidate_block}

You must produce pairwise judgments for ALL of the following pairs:
{pair_block}

Return pairwise comparisons between ALL unique pairs of candidate summaries.
Provide your response in the following jsonl format (remember about the jsonl header!).

IMPORTANT: The "winner" field must be the candidate ID string (e.g. "candidate_0"), NOT
 a numeric index.

```jsonl
{{
"option_0": "candidate_{first_i}",
"option_1": "candidate_{first_j}",
"reasoning": "Short reasoning comparing the two candidates from the decision maker's
perspective.",
"winner": "candidate_xx"
}}
...
```
```

*Prompt 12.* Prompt used in `LILO.get_pair_labels` to infer pairwise preferences between candidate summaries from the accumulated feedback.

## G. Example conversations from the benchmarks

We present example conversations of `LILO` with the DM agent to showcase what are the typical questions that our agent asks and what is the form of feedback that it receives. We would like to highlight how for the Thermal Comfort environment the simulated human responses are more qualitative rather than quantitative in nature.

---

Example conversation (DTLZ2 + piecewise linear)

```
- DM: My goal is to achieve the following thresholds in each outcome: y_1 >= 1.0, y_2
>= 0.8, y_3 >= 0.5, y_4 >= 0.5. Improvements over the thresholds are always good, but
less important than bringing the outcomes to their threshold values. The further away
an outcome is from its threshold, the higher is its negative impact on the overall
performance.
- LILO: How do you prioritize improvements in outcomes that are already above their
threshold values compared to those below their threshold values?
- DM: I prioritize improvements in outcomes that are below their threshold values over
 those that are already above, as meeting these thresholds has a significant impact on
 my overall satisfaction.
- LILO: Are there any interactions or trade-offs between outcomes that you consider
when evaluating overall satisfaction (e.g., is underperforming in one outcome
acceptable if others overperform)?
- DM: Yes, I consider interactions and trade-offs between outcomes, taking into
account their relative importance to my overall satisfaction, which is not solely
determined by their threshold values but by the impact of their improvement on my
satisfaction.
- LILO: Considering outcomes $1_0$ and $1_2$, how do you trade off between $y_1$ being
 slightly above its threshold in $1_0$ and $y_4$ being significantly above its
threshold in $1_2$, given that $y_2$ and $y_3$ are below their thresholds in both
cases? Specifically, which outcome do you prefer and why?
- DM: I prefer outcome $1_0$ because $y_1$ being slightly above its threshold of 1.0
takes precedence due to its higher importance.
- LILO: Considering outcomes $2_0$ and $2_1$, both have $y_1$ above its threshold, but
 $y_2$ is below its threshold in both cases. Given that $y_4$ in $2_0$ is above its
threshold and in $2_1$ is significantly below, and $y_3$ is slightly above its
threshold in both, which outcome do you prefer and why?
- DM: I prefer outcome $2_0$ because it has $y_4$ above its threshold, contributing
positively to my overall satisfaction, unlike $2_1$ where $y_4$ is significantly below
 its threshold.
```

---

*Example 1.* Example fragments of conversation of `LILO` with the DM on the DTLZ2 + piecewise linear environment.

---

**Thermo + type A**

– DM: My goal is to keep all metrics within my thermal comfort preferences.
– LILO: Can you provide more specific thresholds for each metric that you consider to be within your thermal comfort preferences?
– DM: To be honest, I prefer it when the air doesn't feel drafty at my neck, and I don't feel too much warmth coming from the ceiling. I also like it when there's not a big difference in temperature between my head and ankles. As for the floor, I find it comfortable when it's around 22-24 degrees Celsius. If it's too cold or too hot, my feet feel uncomfortable, and that affects my overall comfort.
– LILO: How do you prioritize these metrics relative to one another? For example, is minimizing PPD more important to you than minimizing DR, or are they equally important?

– DM: For me, it's really important that I don't feel drafty or too warm from the ceiling. If I had to choose, I'd say minimizing those feelings is a bit more important than the other factors. But, at the same time, I also value not having a big temperature difference between my head and ankles. It's all about finding a balance, I suppose. If one thing is off, it can really affect how I feel overall.
– LILO: Considering arm_index 1_0 (PPD: 25.67%, DR: 13.52%, dT_vert: 0.24K, dT_pr: 1.02K, T_floor: 18.47C) and arm_index 1_3 (PPD: 9.02%, DR: 3.27%, dT_vert: 4.14K, dT_pr: 17.08K, T_floor: 25.52C), which one do you prefer and why?
– DM: I prefer arm_index 1_3. I feel more comfortable with the conditions in 1_3, even though the temperature difference between my head and ankles is a bit higher. The air feels calmer and less turbulent, which makes me feel more at ease. In 1_0, I feel a bit chilly, especially on the floor, which is slightly too cold for my liking. I wish it were a bit warmer, around 20-22C would be perfect.
– LILO: On a scale of 1-5, how satisfied are you with arm_index 1_5 (PPD: 57.90%, DR: 0%, dT_vert: 0.50K, dT_pr: 19.41K, T_floor: 27.10C), and would you like to see improvements in any specific metrics?
– DM: I'd rate my satisfaction with arm_index 1_5 as a 1 out of 5. I feel quite uncomfortable with these conditions. The air feels stale, and I feel a bit too warm from the ceiling, which is making me feel uneasy. Also, the floor temperature is a bit too high, I prefer it when it's around 20-24C. I'd like to see improvements in reducing the warmth from the ceiling and adjusting the floor temperature to a more comfortable range.

*Example 2.* Example fragments of conversation of `LILO` with the DM on the Thermal Comfort + Type A environment.

