# OpenReview forum: "LILO: Bayesian Optimization with Natural Language Feedback"
_ICML.cc/2026/Conference — ICML 2026 regular_

### Official Review · Reviewer_J6Ni · 2026-02-16

**Soundness:** 3
**Presentation:** 3
**Significance:** 3
**Originality:** 3
**Overall Recommendation:** 4
**Confidence:** 3

**Summary:**

The paper proposes LILO, a framework integrating LLMs with Bayesian Optimization to handle natural language feedback. It uses an LLM to translate free-form feedback into pairwise preferences for a GP surrogate.

**Compliance With Llm Reviewing Policy:**

Affirmed.

**Final Justification:**

The approach of using a capped budget for pairwise comparisons (K) combined with an active selection strategy (qEUBO) and chunked prompting effectively addresses my concerns about quadratic scalability and context window bloat. This successfully clarifies how the computational overhead remains manageable, especially within your targeted regime of expensive, batched experiments.

Because you have adequately resolved my specific technical doubts, I am keeping my positive score (Weak Accept).

However, after following the broader discussion, I also recognize and share Reviewer EU7T's underlying reservations. The lack of controlled, real-world human experiments—and the untested impact of unpredictable human "projection noise" when mapping complex, multi-dimensional preferences into language—remains a tangible limitation that prevents me from raising my score to a full Accept.

**Key Questions For Authors:**

1. The paper compares performance per "trial" (iteration). However, LILO involves heavy LLM inference (question generation, summarization, pairwise labeling). The wall-clock time and computational cost compared to standard BO are significantly higher, yet this trade-off is not analyzed.

2. The experiments are limited to low-dimensional problems (d=4 to 8). GP-based BO is known to struggle in high dimensions.

3. How does LILO handle hallucinated preferences? If the LLM misinterprets a vague user comment as a strict constraint, the GP might be permanently biased against a valid region.

4. The paper compares performance per "trial" (iteration). However, LILO involves heavy LLM inference (question generation, summarization, pairwise labeling). What is the wall-clock time and token cost per iteration compared to standard BO?

**Strengths And Weaknesses:**

## Strengths

- The core idea of bridging the gap between the expressive power of natural language (NL) and the sample efficiency of GP-based BO is well-motivated.
- The separation of concerns—using the LLM for interpretation/labeling and the GP for uncertainty quantification/exploration—is a sound architectural choice. It avoids the hallucination and poor calibration issues often seen in LLM-as-an-optimizer approaches.
- The authors provide extensive ablation studies (prior knowledge, feedback batch size, acquisition functions).
## Weaknesses
- Validating an LLM-based system using an LLM-based user creates a self-fulfilling prophecy. The "DM agent" likely speaks the same distribution of language that the "LILO agent" is trained to understand.
- For a paper claiming to solve "Human-in-the-loop" optimization, the complete absence of real human experiments is unacceptable. Real humans are inconsistent, vague, and irrational in ways LLMs are not.
- The method converts rich NL feedback into binary pairwise labels. This seems like a massive compression and loss of information.

---

> ### Author Rebuttal · Authors · 2026-03-31
>
> We thank the reviewer for their feedback and we address their comments below.
>
> ---
>
> **W1 & W2)** We agree that using LLM feedback introduces the risk that the DM and LILO agents share similar linguistic priors. Our goal in using LLM simulators was to provide a controlled, reproducible environment where the latent utility is known and methods can be compared at scale. We did however take measures to ensure realism of the simulated feedback. We iteratively refined the DM prompts to avoid explicit utility leakage while retaining some vagueness. Our experiments also span multiple semantic contexts and include experiments with different LLMs for LILO and the DM (Appendix E.7), reducing the chance that observed performance arises purely from distribution matching. More broadly, we also note that not all applications of LILO necessarily envision a human in the loop. With the rise of agentic systems, feedback may itself come from another LLM agent endowed with domain knowledge.
>
> We agree that experiments with real humans are an important aspect.  However, a rigorous evaluation would require a dedicated human study with controlled interfaces and large sample sizes. We view this as complementary future work rather than a prerequisite for this paper.
>
> **Practical experience.** Although we do not include a controlled human study, we have deployed variants of LILO in real-world optimization problems involving online experimentation at a large internet firm. Domain experts specified objectives (e.g., improving some engagement metrics without regressing others), while each experimental round required several days of A/B testing. In our experience, NL feedback was easier for experts to provide than repeated pairwise comparisons while enabling LILO to identify promising configurations aligned with the goals. Because these studies involved proprietary systems, long cycles, and limited repeated evaluation, they are difficult to standardize and reproduce as academic benchmarks.
>
> **W3)** We agree that translating NL feedback into pairwise labels is potentially lossy. However, this is a deliberate design choice enabling the use of GP models with calibrated uncertainty estimates. Empirically, we did not find this compression to be a major bottleneck. Figure 4 shows that a single NL message can be as informative as 8–16 noiseless utility evaluations or comparisons. Figure 6 shows that pairwise labeling is more effective than directly asking the LLM for scalar utility scores. Appendix E.4 further shows that LILO predicts pairwise labels with ~85% accuracy. Overall, the results suggest that pairwise comparisons preserve most information relevant for optimization. We do however observe that asymptotically LILO shows diminishing returns, and we hypothesize that in this regime there could be a loss in fidelity of translating NL feedback to pairwise comparisons (see our response to Q1 of reviewer 6Ah4). Yet, our focus is the small-sample regime where this is not an issue.
>
> **Q1 & Q4)** LILO introduces additional computational cost relative to standard PBO. In our framework, the dominant overhead comes from pairwise labeling. Each labeling call takes approximately 30-45 seconds (context-length depending). With a labeling budget of K = 64 and chunk size S = 8, this gives a total runtime of 4-6 minutes per iteration. GP fitting and acquisition optimization take negligible time in comparison. However, the intended setting for LILO is one where each BO trial is itself very expensive and each experimental round may take hours or days. While this overhead is nontrivial relative to PBO, it is still negligible relative to the underlying experiments. Importantly, we found that, LILO made it more natural for experts to provide feedback, increasing their willingness to engage, which is often the real practical bottleneck.
>
>
> **Q2)** We agree that standard GP-based BO becomes challenging in high-dimensional search spaces -- this is a difficult problem albeit orthogonal to the focus of this work. However, LILO is modular with respect to the BO engine and is not tied to standard low-dimensional GP surrogates. In principle, it can be combined with many HDBO advances (e.g., additive kernels, sparse GPs).
>
> **Q3)** We agree that hallucinated preference interpretations may be a potential failure mode. Yet, several aspects of LILO help mitigate this risk: (i) the proxy GP does not treat LLM-derived labels as noise-free, instead it models the uncertainty stemming from inconsistent labels; (ii) the acquisition function continues to explore uncertain regions, making it possible to revisit areas deprioritized earlier (iii) LILO can ask clarifying follow-ups when the DM’s feedback is vague (iv) pairwise labels are re-generated at every trial, making it less likely that a single early misinterpretation dominates throughout the entire optimization process.
>
> ---
>
> We hope these clarifications address your questions and concerns, and we look forward to your feedback.

---

> > ### Author Rebuttal · Reviewer_J6Ni · 2026-04-03
> >
> > Thank you for the clarifications. However, your defense regarding hallucination mitigation raises a new scalability issue.
> >
> > You stated that "pairwise labels are re-generated at every trial" to prevent early misinterpretations from permanently biasing the GP, and noted that labeling currently takes 4-6 minutes per iteration. As the history of experimental outcomes and feedback grows, doesn't re-generating all pairwise labels cause the computational time and context window to scale quadratically? How do you manage this overhead to prevent it from crippling the system in longer optimization campaigns?

---

> > > ### Author Response · Authors · 2026-04-03
> > >
> > > We thank the reviewer for raising this important question. Indeed, in our proposed framework, pairwise labels are re-generated at each BO trial. The reason we re-generate labels is that the system’s belief about the decision maker’s latent preferences is continually updated as new outcomes and new natural-language feedback are observed. Hence, at trial $n$, we want the inferred pairwise preferences to reflect the latest accumulated evidence, rather than remain tied to interpretations made under less informed earlier contexts. Importantly, however, this does not imply that the LLM workload or prompt size grows quadratically with the optimization history.
> > >
> > > First, while the pool of possible pairs over the experimental history grows as $O(|D_n^{\mathrm{exp}}|^2)$, we never enumerate or label all such pairs. Instead, we place a cap of at most $K$ pairwise comparisons per BO iteration and use an active pair-selection strategy based on qEUBO to choose only the most informative pairs under this fixed budget (see Algorithm 2).
> > >
> > > Second, to avoid overloading the context window, labeling proceeds in chunks. Rather than asking the LLM to process all selected pairs at once, each prompt requests labels for $S <= K$ pairs. Thus, even when the optimization history grows, the labeling prompt itself remains fixed-size at the pair level, and the number of LLM calls is simply $\lceil K/S \rceil$. With $K=64$ and $S=8$, as used in our experiments, this yields 8 labeling calls of fixed size per BO iteration. These calls can be made in parallel if needed.
> > >
> > > Thus, the only remaining scalability consideration is the linear, rather than quadratic, growth of the accumulated LILO--DM interaction history, since preference inference is conditioned on the accumulated conversational feedback. In the intended deployment regime of LILO, where BO is used for extremely expensive experimentation and the number of interaction rounds is typically modest, this growth is unlikely to become a practical bottleneck. For substantially longer campaigns, the history could be controlled through summarization or compression of prior feedback, retaining only the most salient information across rounds.
> > >
> > > It is also important to note that our setting is batched: the DM provides feedback once per batched BO round, after $B^{\mathrm{exp}}$ new candidates have already been evaluated. Therefore, after $n$ trials, the system has already observed $B^{\mathrm{exp}} \cdot n$ evaluated candidates. For example, with $B^{\mathrm{exp}}=8$ and a total of $T=8$ trials, the optimization has already evaluated 64 candidates by the end of the campaign, which is comparable to the scale of many practical fully sequential BO runs. In that sense, even a seemingly small number of feedback rounds corresponds to a substantial evaluation budget. Finally, Appendix E.5 further shows that LILO remains competitive even at longer horizon trials – even up to 16 batched trials.
> > >
> > >
> > > Finally, we would like to emphasize that LILO is designed for settings where BO is most valuable: when experiments are expensive and only a few rounds may be realistically feasible (e.g., several hours or days involving wet-lab experimentation, ML training, or large-scale online A/B testing). This is precisely the regime where LILO provides the largest gains over the baselines in our experiments, particularly in the early rounds.
> > >
> > > We hope this clarification resolves the reviewer’s concern regarding the scalability of re-labeling. We will make sure to make this aspect of the paper clear, highlighting that the LLM workload remains explicitly bounded through active pair selection and chunked prompting.
> > >
> > > We thank the reviewer again for the thoughtful comments and we hope our clarifications further strengthen the promise of LILO as a practical way to combine rich natural-language feedback with the sample efficiency of BO. We would be grateful if the reviewer could consider these clarifications when finalizing their assessment.

---

### Official Review · Reviewer_EU7T · 2026-03-11

**Soundness:** 3
**Presentation:** 4
**Significance:** 2
**Originality:** 2
**Overall Recommendation:** 3
**Confidence:** 5

**Summary:**

Authors propose a method that uses natural language feedback to infer pairwise preferences between observed outcomes that in turn is used to fit a pairwise GP surrogate, and apply standard BO acquisition functions to propose the next batch
of candidate.  Authors a propose a simple framework that uses LLM and GP to "approximate" utility estimation of a decision maker.
 One major advantage of LILO is its use of natural language feedback, which can be more information-dense from conventional alternatives. The algorithm begins by estimating/identifying the DM’s high-level optimization goals by generating questions and seeking answers to those.  This is followed by candidate generation and Feedback acquisition.
The key contribution is a proposal of a framework that uses LLMs to parse natural feedback and convert that into an utility estimation engine which can be fed to a BO framework. The broader idea can also be used as a general feedback estimation for other DFL tasks as well.

**Compliance With Llm Reviewing Policy:**

Affirmed.

**Key Questions For Authors:**

-- see above --

**Limitations:**

Authors do specify in conclusion that there are no real world experiments conducted.

**Strengths And Weaknesses:**

The biggest strength of the paper is its fluid and easy to follow writing. Authors make things simple without making it oversimplified. I commend the authors for their clear thinking. Other strength of the work is that the authors have combined two strong ideas in one framework and at least demonstrated through simulations why it should work.  I don't think there is any doubt that LLM as a support to BO would not work, the question is always at what cost and whether the cost is payable ? Further, what are the bottlenecks ? I am afraid thats where the work loses its grip.

I am assuming that the LILO method still requires a human free form feedback (given by the DM) and passes them to the LLM ?  The supporting role that the BO plays is then critically dependent on the quality of the feedback that is available.  One of the criticism of this work according to me are:
1. Scalability of the human feedback :  It is unlikely that any human DM would like to answer many questions. This implies that the biggest challenge with the method would be to perform in a restricted feedback regime while the other methods may not face the same constraint as the complexity of feedback in other methods may not be high.

2. Noisy nature of the feedback: The evaluation metrics provided by a DM are often not easily comparable. The noise associated with natural language feedback can be extremely high, which may actually reduce the expressiveness of the signal and push the system back toward standard pairwise preference models. If the human is inconsistent—liking B over C, but then suddenly liking C over B when A is introduced—the LLM may struggle to distill a stable utility function $g$.

3. Unclear feedback in real world:  Sometimes DM themselves are not clear.  "I like a white Tesla", would not translate to user liking all white cars.  The extra complexity of handling symantic meaning may lead to going in a wrong direction to solve the BO associated with the human utility function ($g$).


4. The experiments were conducted on simulated DM responses. That is actually there are no real-world user feedback --  which to me is the biggest weakness of the work.  Given the applied nature of the work, I think its utility would not be proved unless a real world decision making with humans in the loop is performed which the work lacks.


5. The other weakness to me is concerning the point I wrote above also is that the method is not fit for high dimensional data like images or other data where the human needs to provide a more detailed feedback of features that he may not even derive from the data.  I dont think its a bigger issue, given there are many other areas where this method would work.


6. Authors also noted (E5. page 35) that LILO is only useful when only a few queries/feedback is available vis-a-vis other methods and when the number of feedback increase, the performance of LILO becomes comparable to other preferance BO methods.


There is also a active learning flavor to the work.  I believe authors can take ideas from there as well..

I am more than happy to review the rating again after hearing from the authors on the above feedback.

---

> ### Author Rebuttal · Authors · 2026-03-31
>
> We thank the reviewer for raising the point that the practicality of LILO with humans in the loop depends on the quality of the feedback. We agree that studying the qualitative properties of human feedback is important. However, the primary goal of this paper is different: assuming access to NL feedback, how can it be efficiently used for BO? More broadly, not all applications of LILO necessarily involve a human DM. With the rise of agentic systems, there are increasingly many settings in which feedback may be generated by another LLM agent with domain expertise..
>
> **Our main contribution** is an algorithmic framework that converts free-form language into information modellable with a GP surrogate and thus optimizable with standard BO acquisition functions. We show that this decomposition: LLMs for language interpretation and GPs for uncertainty-aware optimization, substantially outperforms pure LLM-based methods and matches or exceeds baselines that rely on *noiseless* scalar or pairwise feedback.
>
> By contrast, understanding how humans provide language feedback, what is the impact on their cognitive load and how it differs from alternative feedback formats is an important but separate research question. Addressing it rigorously would require a dedicated human study with large sample sizes. We view this as complementary future work rather than a prerequisite for this paper.
>
> **Practical experience.** Although we do not include a controlled human study, we have deployed variants of LILO in real-world optimization settings involving online experimentation at a large internet firm. Domain experts expressed objectives in NL (e.g., improving a subset of engagement metrics without regressing others) and each experimental round required several days of A/B testing. In our experience, NL feedback was easier for experts to provide than repeated pairwise comparisons, while enabling LILO to identify promising configurations aligned with expert goals. Because these studies involved proprietary systems, long experimentation cycles, and limited opportunities for repeated evaluation, they are difficult to standardize and reproduce as academic benchmarks.
>
> **What the framework already addresses.** Our framework and experiments already address several practical concerns:
>
> * **Restricted feedback.** We agree that humans are unlikely to provide long sequences of detailed feedback. LILO is explicitly designed for settings where only a small number of interaction rounds is feasible, which is common in BO problems with expensive experiments. Accordingly, our results show that LILO is most beneficial in the first few trials.
> * **Noise and ambiguity.** NL feedback can be noisy or inconsistent, but this is also true for the alternative scalar ratings and pairwise comparisons. Importantly, the pairwise GP does not assume that LLM-generated labels are noise-free; instead, it explicitly models uncertainty in inferred preferences. In addition, LILO is designed to reduce ambiguity by asking clarifying questions (even before any experimentation has taken place). We also took measures to ensure realism of the simulated feedback. During prompt design for the DM simulator, we iteratively refined the instructions ensuring that the simulator does not explicitly leak the utility function and that the responses retain a degree of vagueness.
> * **High-dimensional outcomes.** LILO does not attempt to model the potentially high-dimensional outcomes Y. Instead, it only models with the GP the composite function from candidate configurations X to scalar utilities. As long as the LLM can interpret the observed outcomes, their format is largely irrelevant. In fact, text or image outputs may be particularly suited for NL feedback, where explicit numerical objectives are often difficult to define while it is relatively easy to provide verbal feedback on the observable texts or images.
>
> **New experiment with text outcomes.** To further illustrate this point, we conducted a new experiment on optimizing an article summarization system. We optimized parameters of an LLM summarizer, treating the generated texts as outcomes. We observed strong performance, matching or exceeding methods with access to the ground-truth utility. Additional details and results can be found at: **https://anonymous.4open.science/r/LILO-ICML-rebuttals-B2AB**.
> In addition, this link also presents **two new benchmark problems** based on the NAS-Bench-201 task illustrating compatibility of LILO with modern applications such as HPO.
>
> **Active learning.** We also appreciate the reviewer’s observation that the method has an active learning flavor. Indeed, our qEUBO-based pair selection for LLM labeling can be viewed as an active acquisition strategy. An interesting future direction is to extend this further by actively selecting the most informative questions for the LILO agent to ask.
>
> ----
> We hope our response fully addresses your concerns and we look forward to hearing your feedback.

---

> > ### Author Rebuttal · Reviewer_EU7T · 2026-04-03
> >
> > I believe lack of real world experiments as also agreed by authors is my biggest concern for the work.  Even though authors mention that they have performed such experiments, but I believe this concern cannot be addressed in this short period .  Other concern was on the sample size.  I stand with my current rating.

---

> > > ### Author Response · Authors · 2026-04-03
> > >
> > > We thank the reviewer for clarifying that the main remaining concern is the absence of controlled real-world human experiments. We agree that this is an important aspect, and we do not claim to have resolved it within the rebuttal period. However, the scope of this paper is narrower: we study the algorithmic question of how free-form natural language feedback can be incorporated into Bayesian optimization in a principled and sample-efficient way. For this reason, we evaluate LILO in a controlled and reproducible setup using a simulated decision maker, which enables extensive ablations and repeated comparisons across environments. In our benchmark studies, all methods were run with 32 independent replications per method–environment setting. The manuscript already explicitly acknowledges the lack of human-subject experiments as a limitation and positions such studies as relevant but complementary future work.
> > >
> > > At the same time, we believe the current evidence is meaningful for the paper’s claimed contributions. The experiments show that natural-language feedback can be more information-dense and can be effectively translated into a numerical signal that is optimizable with standard, principled BO machinery. We also show that LILO is most beneficial in the few-round, feedback-limited regime, where it outperforms conventional baselines. This regime is a key motivation for the method and is especially relevant in real-world settings where each round of experimentation may be costly and time-consuming, such as wet-lab experiments, ML training, or large-scale online A/B testing.
> > >
> > > Regarding the reviewer’s comment about “sample size,” we are unfortunately not fully sure which aspect is intended. If it refers to the number of experimental replications, then each method × environment setting was run with 32 independent replications, which is standard for this type of BO study and provides a reasonably robust basis for method comparison. If it refers to the number of BO trials / interaction rounds, then this is in fact central to our setting: LILO is specifically designed for problems where only a small number of interaction rounds is feasible, and our main experiments therefore focus on short batched BO horizons. If instead it refers to the sample size required for a convincing human-subject study, we agree that such a study would require substantial additional effort, which is precisely why we view it as beyond the scope of the current submission.
> > >
> > > While we understand the reviewer's position, we hope our clarification resolves the sample size concern and makes clear why we view controlled human studies as a complementary next step for this line of work, rather than a prerequisite for establishing the core algorithmic contribution. We kindly ask the reviewer to take these clarifications into account in their final evaluation of the paper.

---

### Official Review · Reviewer_6Ah4 · 2026-03-11

**Soundness:** 4
**Presentation:** 4
**Significance:** 3
**Originality:** 4
**Overall Recommendation:** 5
**Confidence:** 4

**Summary:**

Language-in-the-loop Optimization (LILO) presents a framework for integrating free-form natural language feedback into preferential Bayesian optimization (BO) for improved and faster convergence. The authors utilize the typical BO iterative policy by building a GP proxy model for principled uncertainty calibration, and they tackle the preferential extension via information-dense natural feedback from the decision maker rather than constraining preference elicitation to structured scalar ratings or pairwise comparisons. The key steps include: (1) candidate generation and experimentation via the typical BO acquisition function (based on a surrogate GP proxy model) to accumulate the experimental dataset $D_{n}^{exp}$. (2) feedback acquisition, where the LILO agent generates questions and answers based on $D_{n}^{exp}$ from step 1 and the previously available preference dataset ($D_{n-1}^{pf}$) to obtain an updated preference dataset $D_{n}^{pf}$. (3) utility estimation via LLM labeling + GP modeling, which remains the core of their approach, where the natural language feedback from $D_{n}^{pf}$ is converted into a usable optimization signal by constructing pairwise comparisons ($\mathcal{L}$) between experimental outcomes, labeling $D_{n}^{exp}$ based on $D_{n}^{pf}$ using q-Expected Utility of the Best Option (qEUBO). (4) The pair labels $s_{\mathcal{L}}$ are then used together with $D_{n}^{exp}$ and $D_{n}^{pf}$ to fit the pairwise GP proxy model, which is used in step 1 for the next trial. Across synthetic and real-world benchmarks, LILO reportedly exhibits superior performance in early trials and sustained competitive performance in later ones.

**Compliance With Llm Reviewing Policy:**

Affirmed.

**Final Justification:**

As emphasized in the strengths, the authors arguments against constrained, structured feedback from decision-makers represents an important problem in the extensive practicality of preferential Bayesian optimization (PBO).  Although, the emphasis on the loosely coupled nature of LLM for preference elicitation might seem not principled at first, the authors' methodical approach to still retain a proxy GP model as the inner workhorse of BO remains aligned with the conventional and principled uncertainty calibration of GPs for next-query selection.

My concerns with the diminishing returns of LILO or later catch-up of existing PBO methods in later trials does not take away the innovative use of LLM to tackle the constrained problem settings in PBO. The authors have also done a great job in clarifying the position of their methods with conventional BO approaches. Also, the illustrated real-life deployments (although not reported in paper for proprietary and other reasons as noted by the authors) further demonstrate the potential of this paper and my final score recommendation.

**Key Questions For Authors:**

1. Can the authors comment on the core reason for the diminishing returns of LILO with more trials, as noted in the weaknesses? A note on its resolution would be appreciated.

2. The entry point to LILO requires the agent to generate questions for the decision-makers. How exactly is this done?

3. Can the authors also comment on the feasibility of humans in the loop?

If questions 1 and 3 are sufficiently addressed, I am willing to increase my overall recommendation rating.

**Limitations:**

Yes.

**Strengths And Weaknesses:**

Strengths:
------------------------------------------------------------------------
1. The paper is well motivated, as it presents a clear narrative that addresses existing challenges in preferential BO and LLM-based optimization via in-context learning. For preferential BO, the authors argue against constrained, structured feedback from decision-makers and emphasize the loosely coupled nature of LLM flexibility with the principled uncertainty calibration of GPs. The authors also demonstrate a solid command of the literature.

2. Combining the expressiveness of LLMs with the principled uncertainty calibration of GPs, as stated in (1), presents a viable framework for integrating LLM text representations into BO policy design. This could potentially impact reliable and automated discovery in science and engineering applications amid the growing age of LLMs.

3. The authors present rigorous empirical evaluations to lend credence to the effectiveness of their approach. The synthetic and real-world benchmarks are well explained; the baselines are justified, the reason for the selected acquisition function is also justified, and several ablations are carried out to consolidate their choices.

Weaknesses:
-------------------------------------------------
1. The presented results were not convincingly above existing PBO methods, as both reported BO-based baselines seem to catch up in later trials. This calls into question the superiority of the approach. I must, however, acknowledge that the authors recognized this limitation and provided justification for LILO in the paper by commenting on the value of their approach.

2. Also related to point 1, typical BO trials in real-world settings can run into the hundreds, which further challenges the practicality of this approach within the BO community.

3. The feasibility of their approach with humans as decision-makers remains unclear. Although the authors note this as a possible future research direction, it nonetheless remains a major limitation in real-world settings.

---

> ### Author Rebuttal · Authors · 2026-03-31
>
> We thank the reviewer for their encouraging feedback. Below we address the remaining concerns.
>
> ---
>
> **W1 & W2)** Conventional PBO baselines indeed narrow the gap in later trials. However, LILO is designed for settings where BO is most valuable: when experiments are expensive and only a few rounds may be realistically feasible. This is precisely where LILO is most beneficial, consistently outperforming baselines in the early trials.
>
> Importantly, our goal is not necessarily to outperform conventional PBO in the limit of many trials. Rather, we propose a new interaction paradigm for preference-based optimization that enables a new feedback format — natural language. This is useful because users often struggle to express preferences as equations or consistent ratings, but can communicate them verbally. Moreover, with the rising popularity of agentic AI, NL feedback could equally come from LLM agents. LILO provides an elegant way of leveraging the domain knowledge of specialized agents for BO purposes.
>
> We also emphasize that the baselines in our experiments rely on *noiseless* scalar or pairwise feedback. At larger feedback budgets, these baselines should be viewed as optimistic upper bounds. We also note that our setting is batched: each BO trial evaluates $B^{exp} = d$ candidates in parallel; after only 8 BO trials, the method has already evaluated $8d$ candidates (e.g., 64 candidates for  d = 8). This is comparable to many fully sequential BO campaigns. In applications such as online A/B testing or wet-lab experimentations each round of experiments has high latency. In such cases, a batched approach is necessary to achieve tolerable total optimization runtime. Appendix E.5 further shows that LILO remains competitive even at 16 batched trials.
>
> **W3 and Q3)**  We agree that experimentation with real humans is an important aspect. However, conducting rigorous human studies is a substantial undertaking and would require a dedicated study with large sample sizes (our simulations were run on 32 replicates per each method and environment).
>
> **Practical experience.** That said, we believe LILO is particularly well suited for human-in-the-loop settings. We have deployed variants of LILO in real-world optimization problems involving online experimentation at a large internet firm. In these applications, domain experts specified objectives such as improving certain user engagement metrics without degrading others, while each experimental round required several days of data collection through large-scale A/B testing. In our experience, LILO made it much easier and more natural for experts to provide feedback, increasing their willingness to engage in the process — often a major practical bottleneck — and enabling the method to identify promising configurations aligned with expert goals.
> Because those studies involved proprietary systems, long experimentation cycles, and limited opportunities for repeated evaluation, they are difficult to standardize and reproduce at the level expected for an academic benchmark. We therefore focused the paper on controlled environments where methods can be compared rigorously and reproducibly.
>
> **Q1)** We hypothesize that the diminishing returns of LILO are primarily due to limitations of in-context learning (ICL) in LLMs. Early in optimization, NL feedback is highly valuable: a single sentence can communicate global preferences and trade-offs that would otherwise require many pairwise or scalar labels. This gives LILO a strong advantage off the start.
>
> However, as more observations are collected, conventional PBO methods accumulate enough data to accurately reconstruct the utility function. At the same time, the LLM may saturate as the interaction history grows. ICL performance is known to plateau as the amount of context grows. Appendix E.4. illustrates that although LLMs accuracy in pairwise labeling increases over time it does not reach 100%, it oscillates between 80-90%. We envision several promising directions to mitigate this, e.g. improved context summarization or hybrid approaches combining free-form with structured feedback.
>
> **Q2)** The initial questions are generated with a simple prompt (see Appendix B2, prompt 1). In our simulations, conversations between the DM and LILO are seeded with a high-level message from the DM about their optimization goals (prior to LILO asking any question). These seeding messages are provided in Appendix D.2.
>
> ---
>
> In addition, we added **two new benchmark problems** based on the NAS-Bench-201 task and a **new experiment** on optimizing a text summarization system. These results illustrate that LILO works well in modern applications such as HPO and naturally extends to high-dimensional outputs such as text. We provide the details and results under this link: **https://anonymous.4open.science/r/LILO-ICML-rebuttals-B2AB**.
>
> We hope our response addressed your questions and concerns, and we look forward to your feedback.

---

> > ### Author Rebuttal · Reviewer_6Ah4 · 2026-04-02
> >
> > The authors made a great effort in resolving my concerns and positioning their paper as one that could drive further practical implementations of preferential Bayesian optimization in a less-constrained manner. I have revised my score accordingly.

---

> > > ### Author Response · Authors · 2026-04-03
> > >
> > > We sincerely thank you for your thoughtful re-evaluation and for increasing your score. We greatly appreciate your positive comments on the motivation and positioning of our work, the integration of LLM expressiveness with GP uncertainty calibration, and the rigor of our empirical evaluation. We are grateful that you found our revisions responsive to your concerns and helpful in clarifying the practical potential of our approach. Your feedback is very encouraging, and we are grateful for your support of our work.
> > >
> > > Kind regards,
> > >
> > > The authors of LILO

---

### Official Review · Reviewer_QSCF · 2026-03-13

**Soundness:** 2
**Presentation:** 3
**Significance:** 2
**Originality:** 3
**Overall Recommendation:** 4
**Confidence:** 4

**Summary:**

The paper proposed a Bayesian Optimization (BO) framework, named LILO, that employs a Large Language Model (LLM) to translate free-form natural language feedback and prior knowledge from a decision maker into a structure preference signal. These preferences are then incorporated into a Gaussian Process (GP) model to enable exploration of the optimization problem. Experimental evaluation is conducted on some benchmark problems to demonstrate the effectiveness of the proposed method.

**Compliance With Llm Reviewing Policy:**

Affirmed.

**Final Justification:**

As mentioned in my rebuttal acknowledgement, my concerns have been addressed, so I have decided to increase my score.

**Key Questions For Authors:**

Please address my comments in the Weakness section.

**Limitations:**

The paper does not discuss about the limitations of the proposed method.

**Strengths And Weaknesses:**

Strengths:
+ The paper aims to address preferential BO, which is a good topic.
+ The paper writing is generally clear and easy to follow; I also like the illustrations in the paper.
+ The experiments are described in very detailed.

Weaknesses:
+ The main idea of the proposed method seems to be very simple to me, and I don’t feel it to be interesting. Basically, using an LLM to guide the preference of the decision maker based on available historical preference feedback dataset seems very basic. If we have this type of dataset, then we can make use of many mechanisms to guide the preference. The paper also incorporates prior knowledge to enable warm start of the method, which is also not new to me, as there are works that already demonstrated the use of LLM to enable warm start in BO [1].
+ The experiments are not very comprehensive. Most of the test problems are synthetic. Then the utility functions are pretty simple. I think with these utility functions, even other ML models (not just LLMs) can estimate the preferences well.

[1] Liu, T., Astorga, N., Seedat, N. and van der Schaar, M., 2024. Large language models to enhance Bayesian Optimization. In ICRL 2024.

---

> ### Author Rebuttal · Authors · 2026-03-31
>
> We thank the reviewer for the thoughtful comments and positive feedback. Below we address the remaining concerns.
>
> ---
>
> **W1) Novelty.** We believe there may be a misunderstanding. LILO does not assume access to a historical preference dataset. Instead, the preference data is constructed online during optimization from the DMs’ free-form language feedback. The DM provides their optimization goals and reactions to observed outcomes, and the LLM translates this language feedback into structured preference labels subsequently modeled with a pairwise GP. Ground-truth utilities are never directly observable, they are a latent function of the DM.
>
> While the individual ingredients of our framework are not entirely new in isolation, we believe the novelty lies in their synergetic integration into a BO framework that can operate directly on free-form NL feedback. Existing preferential BO methods typically require structured comparisons or ratings. While these are convenient for optimization algorithms, they can be tedious and cognitively burdensome. In contrast, LILO enables a new feedback format – natural language. We believe this setting to be valuable since many real-world users struggle to specify an explicit utility function, but can easily describe what they like or dislike in language. Moreover, with the rapid development and deployment of agentic AI, NL feedback could equally come from LLM agents. LILO provides an elegant and generic way of leveraging the expertise built into such application-specific agents.
>
> Contrary to most prior works on LLM-for-BO, the LLM in our framework is *not* used as the optimizer itself. Rather, it serves as a semantic parser that converts possibly ambiguous or incomplete language feedback into structured preference information. The optimization is still performed by the GP surrogate and acquisition function, preserving the key strengths of BO: calibrated uncertainty estimates and principled exploration–exploitation tradeoffs. Lastly, our targeted setting is itself novel — the utility function to be optimized is latent and only indirectly observable through free-form language critiques on outcomes of a black-box function. To make the contributions of our work clear we have provided a comparison table with related works under this link: **https://anonymous.4open.science/r/LILO-ICML-rebuttals-B2AB**, which we have now included in the appendix of the paper.
>
> We also acknowledge that using LLMs for warm-starting is not itself new and we do not view it as our primary contribution. It is an optional extension that naturally fits into the LILO framework. Our core contributions remain valid even without the warm-starting module. We will make sure this is made clear in the updated manuscript.
>
> **W2) Experiments.** While several of our benchmark tasks are synthetic, they are not arbitrary toy examples but standard benchmark environments derived from realistic engineering and human-centered design settings. In particular, the Thermal Comfort task is based on the ISO7730 standard while Vehicle Safety and Car Cab Design are derived from engineering simulators.
>
> To further strengthen the realism of the evaluation, we have added **two new benchmark problems** based on NAS-Bench-201 hyperparameter optimization for neural architecture search. We have also added a new experiment on optimizing a text summarization system, illustrating that LILO naturally extends to **high-dimensional outputs such as text**. We provide the details and results under the same link provided above.
>
> Regarding the utility functions, we agree that many are relatively simple — this is often realistic in preferential optimization settings. Prior work has shown that human utility functions are often approximately monotonic, driven by a small number of attributes, additive in nature, and characterized by sparse interactions [1–3]. We therefore chose utility functions that reflect these patterns while still spanning several qualitatively different preference profiles.
>
> Regarding the comment that other ML models could estimate preferences, we would like to clarify that in our setting, the only observable signal is free-form language feedback collected online during optimization. The LLM can generalize from this language feedback and infer pairwise preferences without requiring any task-specific training dataset. Because of this, it is difficult to see how a different conventional ML model could be used with this form of language feedback instead of an LLM, unless additional training data is available.
>
> **Limitations.**
> We would like to kindly note that we do discuss limitations of LILO in Section 6 (please see the last paragraph).
>
> ---
>
> We thank the reviewer again for the helpful feedback. We believe these clarifications and additions strengthen our submission.
>
> **References**
>
> [1] arXiv:2501.18792
>
> [2] doi.org/10.24963/ijcai.2023/421
>
> [3] arXiv:2203.11382

---

> > ### Author Rebuttal · Reviewer_QSCF · 2026-04-03
> >
> > I'd like to thank the authors for spending effort in responding to my concerns. My concerns have been addressed, so I decide to increase my score.

---

> > > ### Author Response · Authors · 2026-04-03
> > >
> > > We sincerely thank you for your thoughtful re-evaluation and for increasing your score. We greatly appreciate your positive comments on the problem's relevance, the clarity of the paper and its presentation, and the detail of the experiments. Your feedback is very encouraging, and we are grateful for your support of our work.
> > >
> > > Kind regards,
> > >
> > > The authors of LILO

---

### Decision · Program_Chairs · 2026-04-30

**Decision:**

Accept (regular)

**Comment:**

The majority of reviewers and the Area Chair agree that this paper introduces a novel and highly promising framework for preferential Bayesian optimization. By leveraging Large Language Models to translate free-form natural language feedback into structured preference signals for a Gaussian Process, the authors present a technically sound and well-written approach. This paradigm seems novel and offers significant potential to lower the cognitive burden on decision-makers compared to traditional structured feedback.

During the discussion, reviewers raised concerns about the empirical evaluation. Specifically, the experiments initially relied heavily on simulated decision-makers and synthetic benchmarks. In response, the authors added new experiments on NAS-Bench-201 hyperparameter optimization and a text summarization task, which addressed the concerns of several reviewers. The method also exhibits diminishing returns in later optimization trials, a limitation the authors acknowledged and attributed to in-context learning saturation.

One reviewer maintains a Weak Reject, raising the concern that human feedback across multiple dimensions is subject to significant "projection noise" that the current framework does not adequately model or address. Two other reviewers share this concern but consider the algorithmic merits sufficient to warrant acceptance. Despite these open empirical questions, the core contribution is original, technically solid, and likely to inspire further work in natural-language-guided optimization. We therefore recommend acceptance.